# Linking diverse nutrient patterns to different water masses within anticyclonic eddies in the upwelling system off Peru

Yonss Saranga José, Heiner Dietze, and Andreas Oschlies

GEOMAR Helmholtz Centre for Ocean Research Kiel

*Correspondence to:* Yonss José (yjose@geomar.de)

**Abstract.** Ocean eddies can both trigger mixing (during their formation and decay), and effectively shield water encompassed from being exchanged with ambient water (throughout their life time). These antagonistic effects of eddies complicate the interpretation of synoptic snapshots as typically obtained by ship-based oceanographic measurement campaigns. Here we use a coupled physical-biogeochemical model to explore biogeochemical dynamics within anticyclonic eddies in the Eastern Tropical South Pacific ocean. The goal is to understand the diverse biogeochemical patterns that have been observed at the subsurface layers of the anticyclonic eddies in this region. Our model results suggest that the diverse subsurface nutrient patterns within eddies are associated with the presence of water masses of different origins at different depths.

## 1 Introduction

Satellite-based measurements have revealed strong correlations between sea surface height and ocean colour (e.g. Cipollini et al., 2001). Most of these correlations are related to nonlinear mesoscale eddies which shape a distinct biogeochemical environment that differs from ambient surrounding waters (e.g. Chelton et al., 2011). Processes proposed to shape distinctly differing environments in energetic mesoscale eddies include eddy vertical transport such as pumping during the formation and intensification (decay) of cyclonic (anticyclonic) eddies (e.g. Jenkins, 1988; Falkowski et al., 1991), and eddy/wind effects driving upwelling in anticyclonic eddies (e.g. Martin and Richards, 2001). Common to these processes is that they have been proposed to enhance near-surface vertical transport and thus increase the availability of essential nutrients in the sun-lit surface ocean (Mahadevan and Archer, 2000; Lévy et al., 2001). Another trait of eddies, somewhat antagonistic, is their role in lateral transport processes. Eddies can enclose water parcels and effectively shield them from being mixed with ambient waters for months (e.g. Dietze et al., 2009). This can render the interpretation of synoptic snapshots, such as obtained by ship-based oceanographic measurement campaigns, problematic because local conditions encountered may well be the result of processes hosted hundreds of miles away in the past rather than being effected locally and contemporarily which, in turn, can obscure causal relationships.

This study sets out to investigate, if and to what extend standing stocks of biogeochemically relevant species are decoupled from local contemporary processes in eddies in the Eastern Tropical South Pacific (*ETSP*). The region is known for oxygen-deprived waters at intermediate depth (Chavez et al., 2008) that host anoxic biogeochemical cycling of organic matter such as denitrification (Codispoti and Christensen, 1985; Farias et al., 2009) and anaerobic ammonium oxidation (anammox) (Hamer-

sley et al., 2007; Lam et al., 2009) – both of which are key to setting the global inventory of bioavailable nitrogen that is essential for phytoplankton growth. A recent survey in the *ETSP* oxygen minimum zone (*OMZ*) revealed strong correlations between mesoscale eddies and subsurface nutrients (Altabet et al., 2012; Stramma et al., 2013). Specifically two anticyclonic eddies were identified associated with both low and high nitrate concentration in their centre. According to the authors of the earlier studies, the nitrate deficit in the low-nitrate eddy was caused by local denitrification and/or dissimilatory nitrate reduction to ammonia (DNRA). As for the high-nitrate eddy, the authors speculated that the nutrient anomaly was related to the initial concentration at the time and place of the eddy formation. We aim here at a more comprehensive analysis of eddies in the region, which explicitly considers their life history. To this end we will apply a coupled, eddy-resolving ocean circulation biogeochemical model of the *ETSP*. Eddies similar to those observed by Altabet et al. (2012) and Stramma et al. (2013) will be analysed. The focus is on nutrient dynamics and water mass properties within the eddies throughout their lifetime.

## 2 Methods

We employ the Regional Oceanic Model System (*ROMS*, Shchepetkin and McWilliams (2005)) to simulate the dynamics of the *ETSP* ocean. This free-surface and terrain-following vertical coordinates model (Shchepetkin and McWilliams, 2005) allows to allocate high resolution in the surface and coastal regions, that are crucial for the biogeochemical processes. The *ROMS* model is coupled with *BioEBUS* (**Bio**geochemical model for the **E**astern **B**oundary **U**pwelling **S**ystems) to simulate the first trophic levels of the food web and the biogeochemical dynamics. Detailed descriptions of the *BioEBUS* is presented in Gutknecht et al. (2013) and in supplementary information of this study (section 1). The first trophic levels of the *BioEBUS* model consist of small (flagellated and microzooplankton) and large (diatoms and mesozooplankton) organisms. The flagellates differ from the diatoms in their adaptation to low nutrients and stratified conditions, as well as their better assimilation efficiency for nutrients. The *BioEBUS* model is a nitrogen-based model that accounts for denitrification, nitrification and anammox. Nitrate, nitrite and ammonium are prognostic variables. The model has been developed to resolve the biogeochemical processes of the eastern boundary upwelling systems under oxic, hypoxic and suboxic conditions.

We employ a 2 ways-nesting procedure (Debreu et al., 2012) to embed a small regional domain within a large-scale domain. The large domain, extends from 69°W to about 120°W in longitude and from 18°N to 40°S in latitude (Fig. SI-1 in supplementary information). It features a horizontal resolution of $\frac{1}{4}^{\circ}$. The embedded domain features a high resolution grid of $\frac{1}{12}^{\circ}$ and extends from around 5°N to 31°S in latitude and around 69°W to 102°W in longitude (Fig. 1 ). Both domains feature 32 vertical levels. The underlying topography of both large-scale and regional domain is derived from the GEBCO at 1 minute resolution (IOC, IHO and BODC, 2003).

The *ROMS* model is forced at the surface with monthly climatological fluxes of heat and fresh water from *COADS* (Worley et al., 2005). Wind fields are from *QuikSCAT* (Liu et al., 1998). At its boundaries, the large scale domain is nudged toward monthly climatological (1990-2010) *SODA* reanalysis (Carton and Giese, 2008). The *BioEBUS* model is constrained at the boundary with monthly climatological nitrate and oxygen from *CARS* (Ridgway et al., 2002). The phytoplankton boundary condition is derived from the monthly climatological surface chlorophyll data from *SeaWiFs* (O'Reilly et al., 1998). This sur-

face chlorophyll data is then extrapolated vertically following the Morel and Berthon (1989) parameterization (Gutknecht et al., 2013). The lateral boundary conditions for zooplankton forcing is based on an analytical function, depending on the vertical distribution of chlorophyll (Gutknecht et al., 2013). For the analysis in this paper, physical and biogeochemical dynamics of the *ETSP* are averaged over 5 years of simulation, after a spin-up time of 25 climatological years.

## 2.1 Model evaluation

Figure 1 shows the simulated physical dynamics of the *ETSP* along with respective observations. At the surface, the simulated eddy kinetic energy (*EKE*) is increased along the coast (Fig. 1-a). This pattern is in agreement with the observed surface *EKE* derived from *AVISO* altimetry product (http://www.aviso.altimetry.fr/duacs/) over a period from 1993 to 2012 (Fig. 1-b). Despite the fact that the model captures the general patterns of surface *EKE*, some differences are notable between the simulated and observed dynamics. These discrepancies might be related to biases both in the *AVISO* data (due to the proximity to the coast, Strub (2001), Saraceno et al. (2008)) and the model solution.

The surface currents, namely the Peru Oceanic Current (POC) in the open ocean and the Peru Coastal Current (*PCC*), transport southern-origin waters north-westward, contributing to the westward South Equatorial Current (*SEC*, contours in Fig. 1-a,b). Along the coast, the model solution shows a southward flow at the subsurface layers (Fig. 1-c,e), in agreement with the observations (Fig. 1-d,f, Chaigneau et al. (2013)). In the north, the southward Equator - Peru Coastal Current (*EPCC*) is intensified in the upper 150 m depth (Fig. 1-c,d). This alongshore current is still present 150 km offshore, with velocities around 5 $\mathrm{cm\,s^{-1}}$. Further south, the simulated subsurface Peru-Chile Under Current (*PCUC*, Fig. 1-e) is deeper (between 50 and 250 m depth) and more intensified than the *EPCC* (Fig. 1-c,d). An equatorward flow inshore is also visible in the simulated dynamics (Fig. 1-e) and is consistent with the observations presented by Thomsen et al. (2016). This equatorward flow is related to the northward PCC observed along the coast ( R. L Smith, 1986; Thomsen et al., 2016). The equatorward current is observed between the surface down to about 60 m depth. Despite the good representation of the alongshore currents, the model underestimates the intensity of both the *EPCC* and the *PCUC*. There are many factors that might have contributed to this discrepancy, among them the low resolution of the boundary conditions used in this simulation and the smoothed bottom topography of the model.

Figure 2 presents the spatial distribution of simulated and observed biogeochemical dynamics. The simulated (Fig. 2-a) as well as the remotely sensed (*MODIS*) surface chlorophyll (Fig. 2-b, Zibordi et al. (2006)) show high concentrations along the coast. The highest values are observed along the Peru upwelling region (between 5°S and 16°S), related to the upward transport of deep nutrient-rich waters (Chavez et al., 2008). Although the representation of patterns of surface chlorophyll is generally good, there are biases offshore where simulated concentrations exceed the observations. We speculate that this model deficiency is related to iron limitation (c.f. Hutchins et al. (2002)), which we do not explicitly account for in our current model. The simulated oxygen content at 300 m depth is relatively high around the equator, as shown in the observed dynamics (Fig. 2-c,d). Vertically, the oxygen distribution shows less oxygenated waters at intermediate depths (Fig. 3-a), in-line with observations (Czeschel et al., 2011, 2015). The simulated nitrate concentrations closely match the observed vertical nitrate distribution, except for the deeper nitrate that is underestimated in the simulated dynamics (Fig. 3-b). The vertical

structure of nitrite concentrations shows high values inshore and in the subsurface layer (Fig. 3-c). This is in agreement with the observations (Fig. 3-c). However, the magnitude of simulated **inner-shelf** nitrite is lower compared to the observations.

## 2.2 The eddy identification

To analyse the source of nutrients within anticyclonic eddies, we identified two anticyclonic eddy types in the simulated
physical-biogeochemical dynamics (Fig. 4 and Fig. 5). Note that the patterns of the selected eddies are consistent with the *in situ* observed biogeochemical patterns presented by Stramma et al. (2013). In this study, Stramma et al. (2013) have two anticyclonic mode water eddies featuring different nutrient patterns at the subsurface layer. One eddy, which was located on the shelf, presented high nitrate and low nitrite in its centre. The second, long lived offshore eddy, exhibited high nitrate and high nitrite concentration in its centre. Once identified, we tracked the eddies on time and analysed both physical and
biogeochemical dynamics within the eddy structure.

The determination of eddy shape is defined based on combination of closed contours of the sea surface height (SSH) anomalies and the Okubo-Weiss parameter (José et al., 2014; Halo et al., 2014). The eddy shape is denoted by the largest connected area inside a closed contour of SSH (Chelton et al., 2011) where vorticity dominates strain (i.e where the Okubo–Weiss parameter is negative, Chelton et al. (2007)). Combining the closed contours and the negative Okubo–Weiss parameter allows to
reduce uncertainties associated with either method (Halo et al., 2014).

### 2.2.1 Subsurface low nitrate and high nitrite eddy ($A_{sim}$)

The simulated eddy $A_{sim}$ is located at the southern part of the Peruvian coast, between 13.5°S and 15.5°S in latitude and 76°W and 78°W in longitude (Fig. 4-a). This eddy features oxygen-depleted intermediate waters (between 100-400 m depth) in its centre, with values below $20\,\mu\,mol\,l^{-1}$ (Fig. 4-b). Nitrate-reduced waters also appear in the centre of the eddy (Fig. 4-c),
apparently indicative of on-going denitrification within the structure, as suggested in previous studies (Altabet et al., 2012; Stramma et al., 2013). At the edge of the structure, the nitrate concentration is higher compared to the concentrations in the centre of the eddy (Fig. 4-c). This high nitrate at the edge might be entrained from the surrounding environment by horizontal stirring. In contrast, the nitrite concentration is higher in the centre and low at the edge (Fig. 4-d).

Generated in the southern part of the Peruvian shelf (around 14.5°S) about 42 days before the instant presented in Figure 4,
the eddy $A_{sim}$ propagates north-westward. This eddy genesis and propagation is in agreement with altimetry observations (Chaigneau et al., 2008). The eddy $A_{sim}$ is intensified from the surface down to 500 m depth, with maximum velocity above $30\,cm\,s^{-1}$ (Fig. 4-e,f). Further, the eddy $A_{sim}$ presents a subsurface velocity maxima, similar to the observed patterns within anticyclonic eddies in the *ETSP* (Chaigneau et al., 2011; Holte et al., 2013; Stramma et al., 2013; Thomsen et al., 2016). This characteristic is related to the poleward flowing *PCUC* (Chaigneau et al., 2011; Colas et al., 2012; Holte et al., 2013).
Temperature and salinity show depressed isolines in the centre of the eddy (Fig. 4-g,h), typical of a mode-water anticyclonic eddy.

### 2.2.2 Subsurface high nitrate and nitrite eddy ($B_{sim}$)

The simulated eddy $B_{sim}$ is an open-ocean eddy, located between 12°S and 14.5°S in latitude and 84°W and 88°W in longitude (Fig. 5-a). This eddy also presents extremely low oxygen concentrations at intermediate depths in its center (Fig. 5-b), which is similar to eddy $A_{sim}$. The oxygen-depleted waters are enclosed by well-oxygenated waters at the eddy's edge (Fig. 5-b). The eddy $B_{sim}$ shows increased subsurface nitrate concentrations in the centre and slightly reduced nitrate concentrations along the edge (Fig. 5-c). The vertical structure of nitrite concentrations shows isolated subsurface high-nitrite waters in the centre of the eddy (Fig. 5-d). Despite reproducing the subsurface nitrite maxima, the simulated concentrations are an order of magnitude lower compared to the ***in situ*** observations of nitrite concentration within eddy (Stramma et al., 2013).

The age of eddy $B_{sim}$ is about two months (54 days) and it was generated offshore near to 85°W and 12°S. The place of generation of model eddy $B_{sim}$ is in agreement with the eddy genesis inferred from the altimetry observations (Chaigneau et al., 2008). Possibly detached from a meander type structure, the eddy $B_{sim}$ propagates westward and is deflected poleward. Figure 5 -e and Figure 5 -f show the vertical section of the eddy $B_{sim}$ velocities. Maximum velocities are found at the surface layers in the westward flow and at the subsurface layers in the eastward flow, respectively. This circulation pattern is similar to the observed velocity within anticyclonic eddies in this region (Chaigneau et al., 2011; Holte et al., 2013; Stramma et al., 2013; Thomsen et al., 2016) and is likely linked to the dynamics of the *PCUC* (Chaigneau et al., 2011; Colas et al., 2012; Holte et al., 2013). The eddy $B_{sim}$ shows elevated temperature and salinity in the centre, characteristics of a mode-water anticyclonic eddy (Fig. 5-g,h). The warmer and saltier waters within the mode water eddies are related to the depression of the lower isotherms and isohalines in the interior of the structure.

### 2.3 Analysis of physical and biogeochemical dynamics within the eddy

In order to have insights into the processes controlling the nutrient distribution within the eddies, the advective transport (vertical and horizontal), sources and sinks of nutrients **(referred in the text as the local biogeochemical processes)** are analysed during the eddy evolution. The time evolution of nutrient concentrations is also presented. The eddy volume is defined here as the volume between 100 m and 200 m within the eddy shape **(Fig. SI-2 in supplementary information for further details)**. The horizontal advective transport of nutrients is calculated at the edge of the eddy structure, while the vertical transport is calculated at the upper (100 m) and lower (200 m) extremes of the eddy volume. The nutrient concentration, as well as the nutrient sources and sinks correspond to the averaged quantities within the eddy volume. The sources and sinks of nitrate consist of nitrification and denitrification, respectively. For nitrite, multiple sources and sinks are accounted. The sources are denitrification and nitrification, while the sinks consist of nitrification, denitrification and anammox.

### 2.4 Particle release experiment

In order to have a more general overview of the processes controlling the dynamics of the eddies in the *ETSP*, we conduct a particle-release experiments and analyse the anticyclonic eddies that are in the vicinity of the particle-release locations at the time of the release. In these experiments, particles are released in three different locations in *ETSP*: (1) along the shelf between

3°S - 15°S, (2) along the shelf between 9°S - 11°S and (3) offshore between 13°S -15°S in latitude and 85°W-86°W (Fig. SI-3 in supplementary information). The particles are released in the entire water column on the shelf and in the upper 300 m at the offshore site, in early austral summer (January) and early winter (June) of the last three climatological years of the model simulation.

## 3 Results

### 3.1 Local biogeochemical dynamics within the eddies

In order to assess the contribution of local biogeochemical dynamics to nutrient variations within the eddy, we analyse here the production and reduction rates of each nutrient at the instant shown in Fig. 4 and Fig. 5. Note that the magnitude of simulated nutrient production and reduction rates, which varies from 0.01 to 1 $\mu\,mol\,l^{-1}, d^{-1}$ ( Fig. 6 to Fig. 9), are in the range of the observed denitrification, nitrification and anammox rates in the upwelling system off Peru (Ward et al., 2009; Kalvelage et al., 2013). Figure 6 shows the vertical distribution of reduction and production rates of nitrate (Fig. 6-a-b) and nitrite (Fig. 6-d-e) within the eddy $A_{sim}$. The net production (production - reduction) is also presented, for both nitrate (Fig. 6-c) and nitrite (Fig. 6-f). Below the surface of the eddy $A_{sim}$, the nitrate reduction by denitrification (Fig. 6-a) is lower when compared to the production by nitrification (Fig. 6-b), showing a net nitrate increase due to biogeochemical processes within the eddy (Fig. 6-c). Regarding the nitrite dynamics, production (by nitrification and denitrification) and reduction (by nitrification, denitrification and anammox) rates show similar patterns. Below the surface, the nitrite reduction is slightly higher than the production (Fig. 6-d-f), suggesting that local biogeochemical processes can not explain the accumulation of high nitrite values in the centre of this eddy (Fig. 4-d). Figure 7 shows the time evolution of production and reduction rates of both nutrients (nitrate and nitrite) within the eddy $A_{sim}$. For comparison, nutrient concentrations during this period are also presented. Both reduction and production of nutrients within the eddy do not show any trend that could explain the reduced nitrate in the centre of the eddy. The nitrate production is higher than the reduction, showing a net increase of nitrate due to biogeochemical processes within the eddy $A_{sim}$ (Fig. 7-a). But the magnitude of the locally-produced nitrate is low compared to total changes in nitrate concentration during the eddy propagation. Local nitrite reduction is higher than local production during the eddy propagation (Fig. 7-b). The pattern of local net nitrite production is not correlated with the total changes in nitrite concentrations. These results show that the low nitrate (high nitrite) within the eddy $A_{sim}$ is not primarily controlled by local biogeochemical dynamics within the eddy $A_{sim}$.

In the eddy $B_{sim}$, the vertical structure of production and reduction rates of nitrate show a net nitrate production in the upper layers (Fig. 8-a,b,c). The local nitrate dynamics during the propagation of eddy $B_{sim}$ show elevated nitrate production in comparison to the reduction (Fig. 9-a). Despite the net increase, the pattern as well as the magnitude of the locally produced nitrate do not match with changes in total nitrate concentrations. The vertical structure of nitrite production also reflects elevated values in the upper surface layer (Fig. 8-d,e,f). Below the surface, nitrite reduction dominates the local dynamics, with no difference between the centre and the edge of the eddy. The nitrite dynamics during the eddy propagation show a local

dominance of nitrite production within the eddy, which is inconsistent with the temporal variations of nitrite concentrations (Fig. 9-b).

The inconsistency between local biogeochemical sources and sinks of nitrate and nitrite and total temporal changes of those nutrients within the eddies suggest that the biogeochemical dynamics do not exert the dominant control on the variation in nutrient concentrations.

## 3.2    Nutrient transport and water mass properties within the eddies

One important aspect of eddies is their capacity to entrain surrounding waters during their propagation (José et al., 2014). These dynamics have been pointed to stimulate the productivity within anticyclonic eddies (José et al., 2014). In the eastern tropical oceans, **anticyclonic** eddies have been observed to carry low-oxygen waters out of the core region of the *OMZ* (Stramma et al., 2014; Karstensen et al., 2015). This makes the analysis of the eddy life history essential for understanding the dynamics in the eddy interior. To investigate the origin of water masses present in the selected eddies, we analyse the advective transports of both nitrate and nitrite into the eddy during the eddy's lifetime (Fig. 10 and Fig. 12). The water mass properties within the structure are also analysed and compared with the surrounding environment during different instants of the eddy's lifetime (Fig. 11 and Fig. 13). Figure 10 illustrates the nitrate and nitrite fluxes into the eddy $A_{sim}$. It shows a strong injection of nutrients from the lateral margins of the eddy. This nutrient injection is elevated in the first months following the eddy formation. The cumulative fluxes of both nitrate and nitrite significantly increase in this period and follow the evolution of both nitrate and nitrite within the eddy. These dynamics suggest a strong exchange with the surrounding environment during this period. This is also visible in the water mass properties within the eddy structure (Fig. 11). At the surface, waters present within the eddy $A_{sim}$ are relatively cool and fresh compared to the water masses present following the eddy formation (Fig. 11 -a-b). These waters are distinct from those from the shelf. Waters in the eddy interior show a dominance of relatively warmer and saltier surface waters compared to the shelf waters (Fig. 11-a-b). When the eddy $A_{sim}$ was formed, the surface waters were even warmer and saltier, characteristic of offshore waters (Fig. SI- in supplementary information). During the propagation of eddy $A_{sim}$, the water masses within the structure are mixed with colder and fresher coastal waters, resulting in heat and salt loss (Fig. 11-c). However, water masses within the eddy are still dominantly from the offshore region. The offshore region is a nitrate-poor environment, with minimum concentrations below $5\,\mu\mathrm{mol\,l^{-1}}$ (Fig. 11-d). At 100 m depth, water masses within the eddy are relatively fresher and cooler and do not match those of the offshore environment (Fig. 11-e,f). Although roughly associated with shelf waters, the water masses within the eddy at this depth are saltier and warmer than those from the shelf. At eddy formation, the water masses within the eddy were even warmer and saltier (Fig. 11-g), suggesting a dominance of offshore waters within the structure. This suggestion is supported by the dominance of warmer and saltier offshore waters in the eddy centre right after the eddy formation (Fig. SI-4-e,f in supplementary information). During the eddy propagation, the fresher and colder shelf waters entered the eddy interior, resulting in a cooling and freshening of the water masses within the structure. As a result, shelf waters carrying high nitrate concentrations dominate the eddy interior (Fig. 11-h). Further down (at 200 m depth), warmer and saltier shelf waters dominate the properties within the eddy$A_{sim}$ (Fig. 11-i,j). **At the eddy formation, both shelf and offshore waters occupy the eddy interior (Fig. SI-4-i-k). The surrounding offshore waters**

have higher concentrations of both nitrate and nitrite (Fig. SI-4-k,l). With the eddy propagation, water properties are modified (Fig. 11-k). Shelf waters, which have lower nitrate concentrations compared to the offshore environment, are kept in the eddy centre and advected during the eddy propagation (Fig. 11-l). Thus, the nitrate concentrations in the eddy interior are lower than those of the surrounding waters, even though interior concentrations show an increase over time during the propagation of the eddy.

The nutrient fluxes across the edge of the eddy $B_{sim}$ are presented in Figure 12. It shows a contribution of both horizontal and vertical transport to the nutrient variation within the eddy, during the eddy's lifetime. After the eddy $B_{sim}$ formation, the nitrate fluxes through the edge of the eddy $B_{sim}$ are dominantly out-going, showing a loss of nitrate to the surrounding environment (Fig. 12-a). These out-going fluxes reduce the nitrate availability within the eddy. About half a month later, the nitrate concentration within the eddy increases. This increase is to a large extent due to the nitrate supply into the eddy structure from both vertical and horizontal boundaries. On the contrary, the nitrite supply into the eddy is largest and positive in the month following the eddy formation and decreases afterwards (Fig. 12-b). Located far from the shelf region, the water masses within the eddy $B_{sim}$ are distinct of those from the shelf (Fig. 13). At the surface, relatively cooler and saltier waters occupy the eddy interior (Fig. 13-a,b). These waters result from a mixture of saltier and cooler southern with fresher and warmer northern waters. During the eddy propagation, intrusion of warmer northern waters increases the eddy surface water temperature (Fig. 13-c). This intrusion of warmer waters can also be seen from the limb of warmer waters at the north-western side of the eddy (Fig. 13-b). At 100 m depth, warmer and saltier southern waters occupy the eddy interior (Fig. 13-e,f,g). These waters also dominate the eddy interior at the instant following the eddy formation (Fig. 11-e-f). The waters of southern origin that are trapped within the eddy's interior are subsequently mixed with the fresher and colder northern waters, which enter the eddy by the north-western edge. However, only relatively small changes occur during the eddy propagation (Fig. 13-g). Consequently, nitrate-poor southern waters are trapped within the eddy (Fig. 13-h). The weak water mass exchange with surrounding waters during the eddy propagation is also visible at 250 m depth (Fig. 13-k). Further down, warmer and saltier northern waters dominate the eddy interior (Fig. 13-i,j). These waters are rich in nitrate (Fig. 13-k), explaining the elevated nitrate in Figure 5. These water mass properties and nutrient distribution are similar to those encountered within the newly formed eddy (Fig. SI-5-i-l in supplementary information).

### 3.3 Eddy stirring and nutrient entrainment

The eddies $A_{sim}$ and $B_{sim}$ showcase that the nutrient supply by physical dynamics is the dominant mechanism that controls simulated (diverse) nutrient pattern within the eddies. The nutrient exchange with surrounding waters occurs throughout the entire lifetime of the eddies. This indicates that the nutrient availability in the vicinity of the eddy plays a role for the nutrient distribution within the eddy's structure. To elucidate this suggestion, we carried out particle-release experiments (subsection 2.4) and analysed the eddies that passed and/or were generated close to the particle-released areas. Figure 14 illustrates the particle distribution in the subsurface interior (between 100 - 200 m depth) of anticyclonic eddies during their propagation. From the early stages on, particles are entrained and trapped within the eddy structures. These particles are transported offshore during propagation of the eddies. Every tracked eddy shows a pronounced temporal variation of the amount of particles within

the structure, an indicative of exchange of properties with surrounding waters. This behavior occurs in eddies tracked during both austral summer (Fig. 14-a,c,e ) and austral winter (Fig. 14-b,d,f ).

## 4 Discussion

Subsurface anticyclonic eddies, also known as mode-water eddies, are common features in the *ETSP* (Chaigneau et al., 2011; Holte et al., 2013). These eddies have a weak surface signal, making it difficult to observe them from space. Based on model results, Colas et al. (2012) have found that these eddies dominate the subsurface of the *ETSP*. They are detached from the subsurface poleward undercurrent *PCUC* (Colas et al., 2012). Only recently, measurements along the Peruvian shelf have shown a direct link between *PCUC* and the mode water eddies (Thomsen et al., 2016). These mode-water anticyclonic eddies propagate westward and transport warm and salty equatorial subsurface in their centre (Chaigneau et al., 2011; Holte et al., 2013). Consequently, these mode water eddies impact the water mass and biogeochemical property distributions of this region. Recent ship-based oceanographic measurements revealed the existence of diverse nutrient patterns within those eddies (Altabet et al., 2012; Stramma et al., 2013). A surprising finding was that of two observed anticyclonic eddies one had low and one had high nitrate concentrations at their subsurface layer (Stramma et al., 2013). The processes behind this diversity are discussed controversially. Altabet et al. (2012) and Stramma et al. (2013) suggest that nitrate consumption by denitrification is the cause of the low nitrate in one of the observed anticyclonic eddies. Their suggestion is supported by the high nitrite concentration measured within that eddy. In contrast, Thomsen et al. (2016) suggests that the low nitrate (high nitrite) water was entrained from the shelf region and trapped within the eddy structure rather than being produced locally by on-going denitrification. For the high nitrate eddy, Stramma et al. (2013) suggest the conditions at the eddy formation to play a significant role on the observed nutrient pattern. Using a coupled physical-biogeochemical model, we aim to understand the processes responsible for the observed different nutrient patterns within anticyclonic eddies off Peru. Our approach here is to assess local production and consumption rates relative to physical exchange of both nitrate and nitrite in the eddy interior with the environment outside the eddy. This is done for an extended period in two simulated anticyclonic eddies which feature nutrient patterns similar **to** the observed patterns. In the low nitrate (high nitrite) $A_{sim}$ eddy (Fig. 4-c), the on-going nitrate reduction by denitrification is lower than the nitrate production by nitrification. This fails to explain the low subsurface nitrate within the eddy, which is in contrast to the interpretation of the observations by Altabet et al. (2012) and Stramma et al. (2013). Further, we find in our simulation that the advective fluxes across the edge of the eddy $A_{sim}$ shows a strong nitrate supply into the eddy, in the first days prior to the eddy formation. This nitrate supply, which is predominately horizontal, is a consequence of the exchange of water masses with the surrounding environment during the eddy propagation. As for the nitrite dynamics, there is a nitrite supply into the eddy time subsequent to the eddy formation. This nitrite supply is higher than the biogeochemically produced concentrations. This result supports Thomsen et al. (2016) who suggest that the low-nitrate (high nitrite) waters were entrained from the shelf region and trapped within the eddy interior. The analysis of water mass properties at three different layers of the eddy $A_{sim}$'s interior shows a presence of water masses of different origin at different depths. At the surface, water masses are predominantly originating from the offshore region, characterized by relatively high temperatures and low nitrate

concentrations. The low nitrate water at the subsurface layer originates from the shelf environment. These distinct water masses are trapped within the eddy structure. During the eddy propagation, there is an exchange of water masses with the surrounding environment, resulting in a reduction of temperature up to $4^\circ$C from the eddy formation to the instant presented in Figure 4. The dynamics within the high-nutrient eddy $B_{sim}$(Fig. 5) also show exchange of water masses with the surrounding environment.

The nitrate variation within the eddy appears to be related to the advective fluxes, which are an order of magnitude higher than the local production and consumption rates. The analysis of the nitrite dynamics within the eddy $B_{sim}$ shows some important aspect: (1) advective fluxes are of a similar magnitude as are the biogeochemical fluxes, (2) the nitrite availability outside the eddy is low during and after the eddy formation (Fig. SI-5 in supplementary information) and during its propagation (Fig. 13). These results suggest that the effect of biogeochemical processes on nutrient patterns within the eddy is important when

the nutrient supply by physical processes is weak. However, the simulated local biogeochemical fluxes are not large enough (order of $1 \mathrm{n}\,\mathrm{mol}\,\mathrm{l}^{-1}\,\mathrm{d}^{-1}$) to produce significant changes in nutrient concentrations. The nutrient fluxes into the eddy during its propagation depend on the nutrient concentrations in the environment surrounding the eddy. This argument also applies to eddy $A_{sim}$: The nitrite-rich waters were entrained from the shelf and injected into the eddy $A_{sim}$ during its propagation. Anticyclonic eddies tracked during the particle-release experiments corroborate this suggestion and show the occurrence of

water mass exchange between the eddy and the surrounding environment. Particle numbers within these eddies are repeatedly increased and decreased, showing a loss and gain of quantities to/from the surrounding environment.

## 5   Conclusions

In this study we used a simulated bio-physical dynamics to investigate the processes responsible for the observed diverse nutrient patterns within two anticyclonic eddies off Peru. Two anticyclonic eddies, based on their subsurface nitrate and nitrite

patterns, were selected from the simulated dynamics and analysed. The model results show a decoupling between local nitrate reduction (nitrite production) via biogeochemical processes and total changes in nitrate (nitrite) within the eddy. Further, they indicate that the advection processes at the edge of the eddy plays an important role in controlling the nutrient variability within the structure. In addition, the analysis of water mass properties show that the nutrient signature within the selected structures is related to the presence of water masses from different origins. In a more general context the particle-release

experiments realized in this study also enhance the role of water mass exchange between the eddies and the surroundings on setting the properties within the eddy structure.Our findings suggest that the biogeochemical patterns at the subsurface layer of the observed eddies in the *ETSP* are likely to be related to the presence of water masses from different origin, which are trapped and are retained within the structure.

*Acknowledgements.* This work is financially supported by the Deutsche Forschungsgemeinschaft (DFG), under the Sonderforschungsbere-

ich 754 "Climate-Biogeochemistry Interactions in the Tropical Ocean" project (www.sfb754.de). Simulations were performed using the computing facilities of the Christian-Albrechts-Universität zu Kiel (NESH) and Norddeutscher Verbund zur Förderung des Hoch- und Höchstleistungsrechnens - HLRN. *In situ* circulation data were provided by the Instituto del Mar del Perú (IMARPE). We are grateful to A.

Chaigneau for providing the *in situ* observations. Nutrient and oxygen *in situ* data were obtained from the German Collaborative Research Center (SFB 754).

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

**Figure 1.** Annual mean surface height (contour every 6 cm) overlaid on eddy kinetic energy ($[cm^{-2}\,s^{-2}]$, colour), corresponding to: (a) model simulation and (b) *AVISO* observation. Vertical distribution of simulated (c, e) and observed (d, f) mean alongshore velocity $[cm.s^{-1}]$, averaged over 3-6°S (c, d) and at around 12°S (e, f). The black lines in (c, e) are the upper and onshore limits of the observed velocities. The observed velocities in Figure 1-d,f were provided by IMARPE and described in detail by Chaigneau et al. (2013).

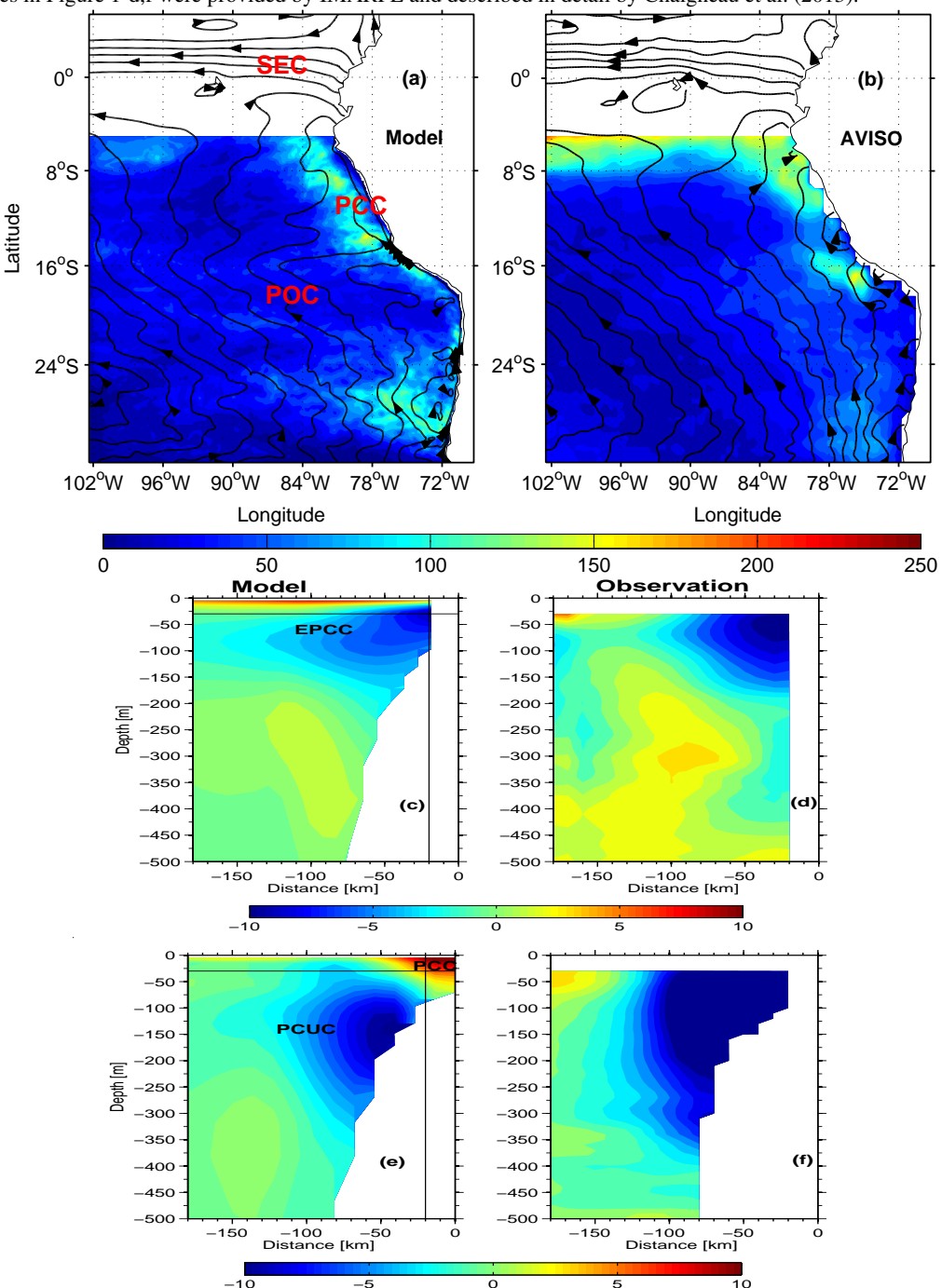

**Figure 2.** Sea surface chlorophyll concentration (in $\log_{10}$, $[\mathrm{mg\,m^{-3}}]$) corresponding to (a) model simulation and (b) *MODIS* observation. Oxygen concentration at 300 m depth $[\mu\,\mathrm{mol\,l^{-1}}]$ corresponding to (c) model simulation and (d) *CARS* observation.

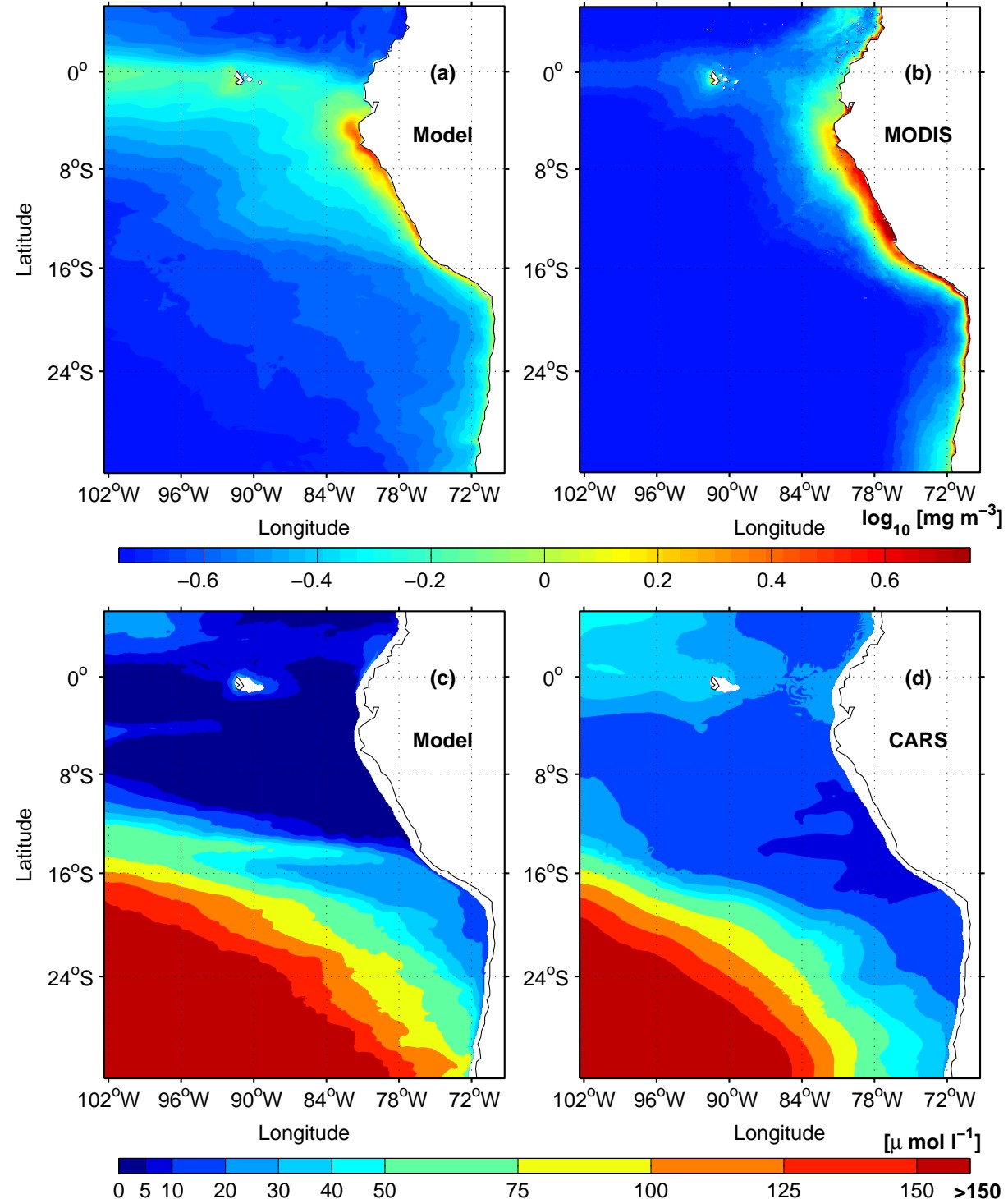

**Figure 3.** Vertical section of oxygen ([μ mol l$^{-1}$], a), nitrate ([μ mol l$^{-1}$], b) and nitrite ([μ mol l$^{-1}$], c) concentrations along 12 °S. Simulated dynamics correspond to climatological December. The observed dynamics is based on measurements from the cruise M91, December 2012. Details about this cruise are described in Czeschel et al. (2015).

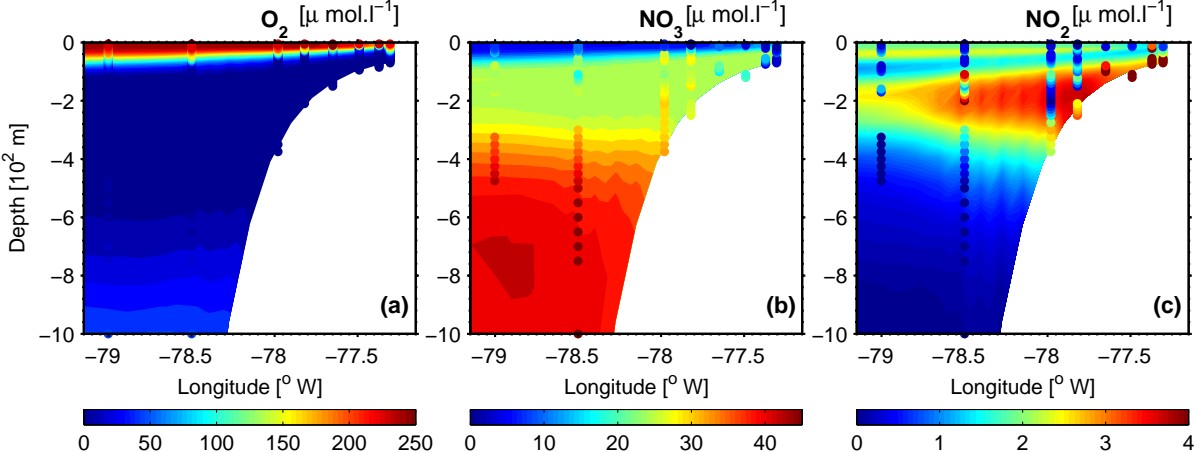

**Figure 4.** (a) Sea surface height [cm], (b) oxygen [μ mol l$^{-1}$], (c) nitrate [μ mol l$^{-1}$], (d) nitrite [μ mol l$^{-1}$] concentrations, (e) meridional velocities [cm s$^{-1}$], (f) zonal velocities [cm s$^{-1}$], (g) temperature [°C] and (h) salinity across the eddy A$_{sim}$ center. Nitrate, oxygen, nitrite, temperature, salinity and meridional velocity section around 14.75°S (magenta full line in Fig 4-a). Section for zonal velocity around 77.65°W (magenta dashed line in Fig 4-a). This dynamics corresponds to 21 July model year 30. The temporal evolution of the eddy is indicated by the red dots.

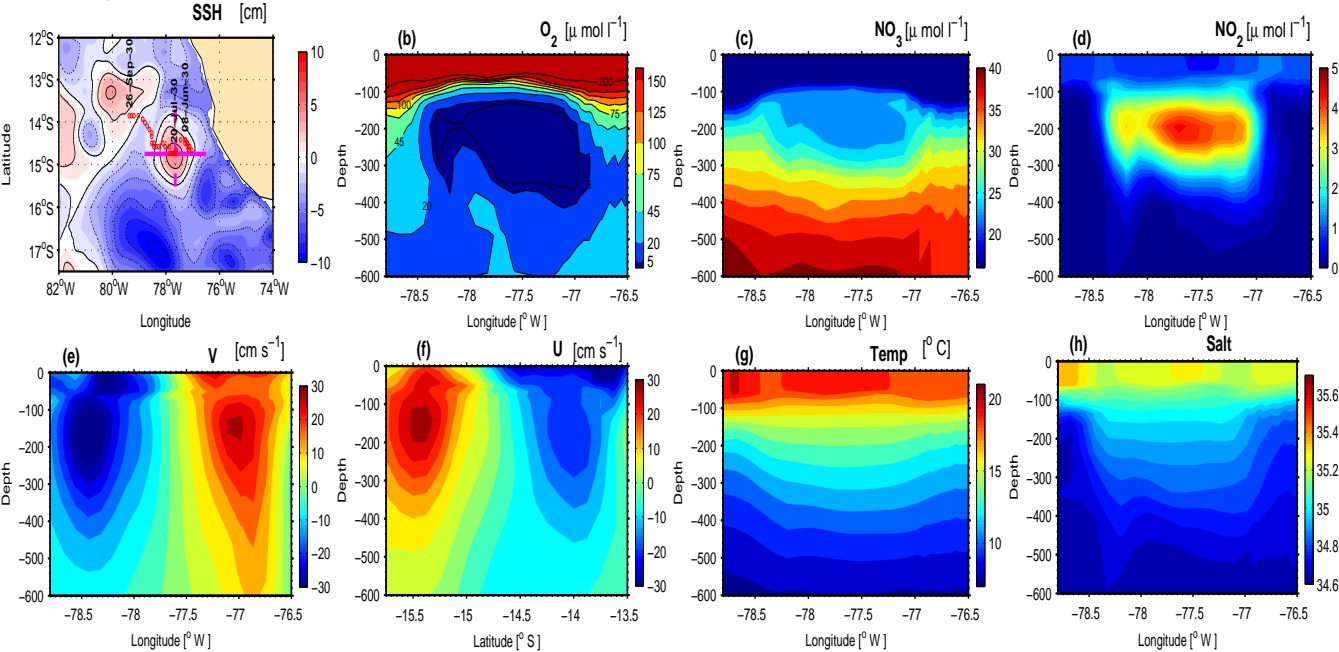

**Figure 5.** (a) Sea surface height [cm], (b) oxygen [µ mol l$^{-1}$], (c) nitrate [µ mol l$^{-1}$], (d) nitrite [µ mol l$^{-1}$] concentrations, **(e)** meridional velocities [cm s$^{-1}$], (f) zonal velocities [cm s$^{-1}$], (g) temperature [°C] and (h) salinity across the eddy B$_{sim}$ center. Nitrate, oxygen, nitrite, temperature, salinity and meridional velocity section around 13.2°S (magenta full line in Fig 5-a). Section for zonal velocity around 85.8°W (magenta dashed line in Fig 5-a). This dynamics corresponds to 27 January model year 28.The temporal evolution of the eddy is indicated by the red dots.

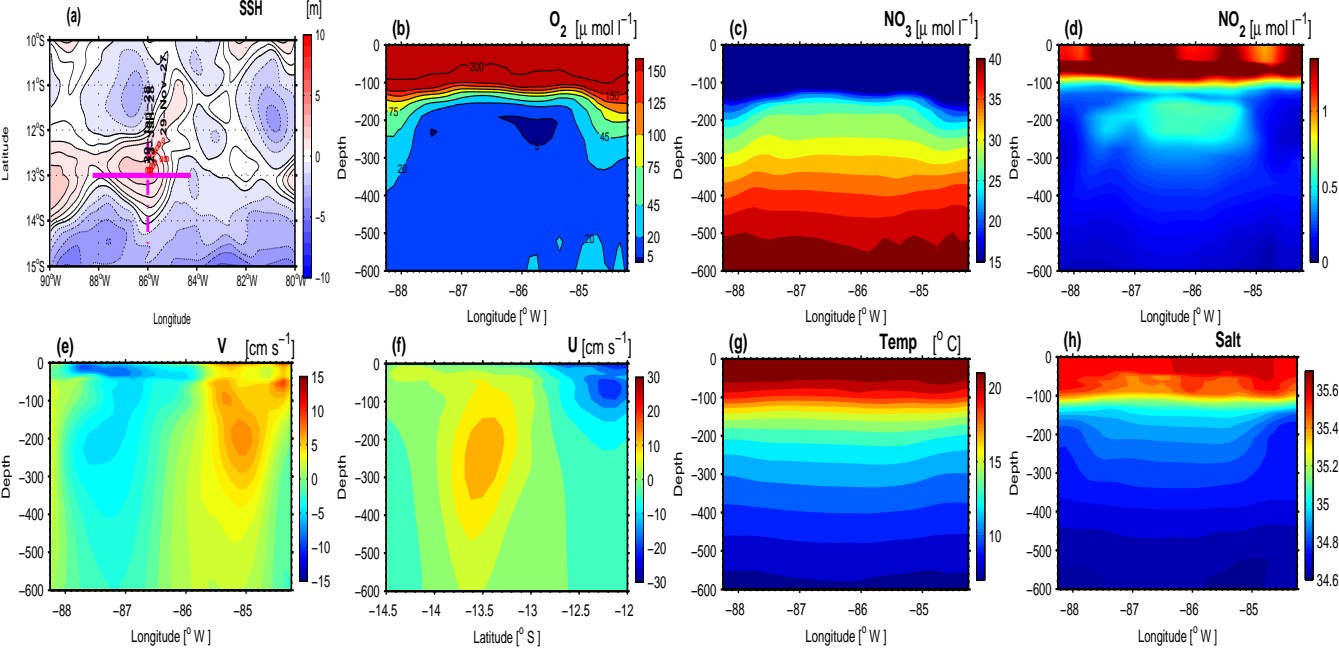

**Figure 6.** Eddy $A_{sim}$ vertical structure of: (a) nitrate reduction by denitrification [$\mu\,mol\,l^{-1}\,d^{-1}$], (b) nitrate production by nitrification [$\mu\,mol\,l^{-1}\,d^{-1}$], (c) nitrate production-reduction difference [$\mu\,mol\,l^{-1}\,d^{-1}$], (d) nitrite reduction (nitrification + denitrification + anammox, [$\mu\,mol\,l^{-1}\,d^{-1}$]), (e) nitrite production (denitrification + nitrification, [$\mu\,mol\,l^{-1}\,d^{-1}$]) and (f) nitrite production-reduction difference [$\mu\,mol\,l^{-1}\,d^{-1}$] at the instant shown in Fig.4.

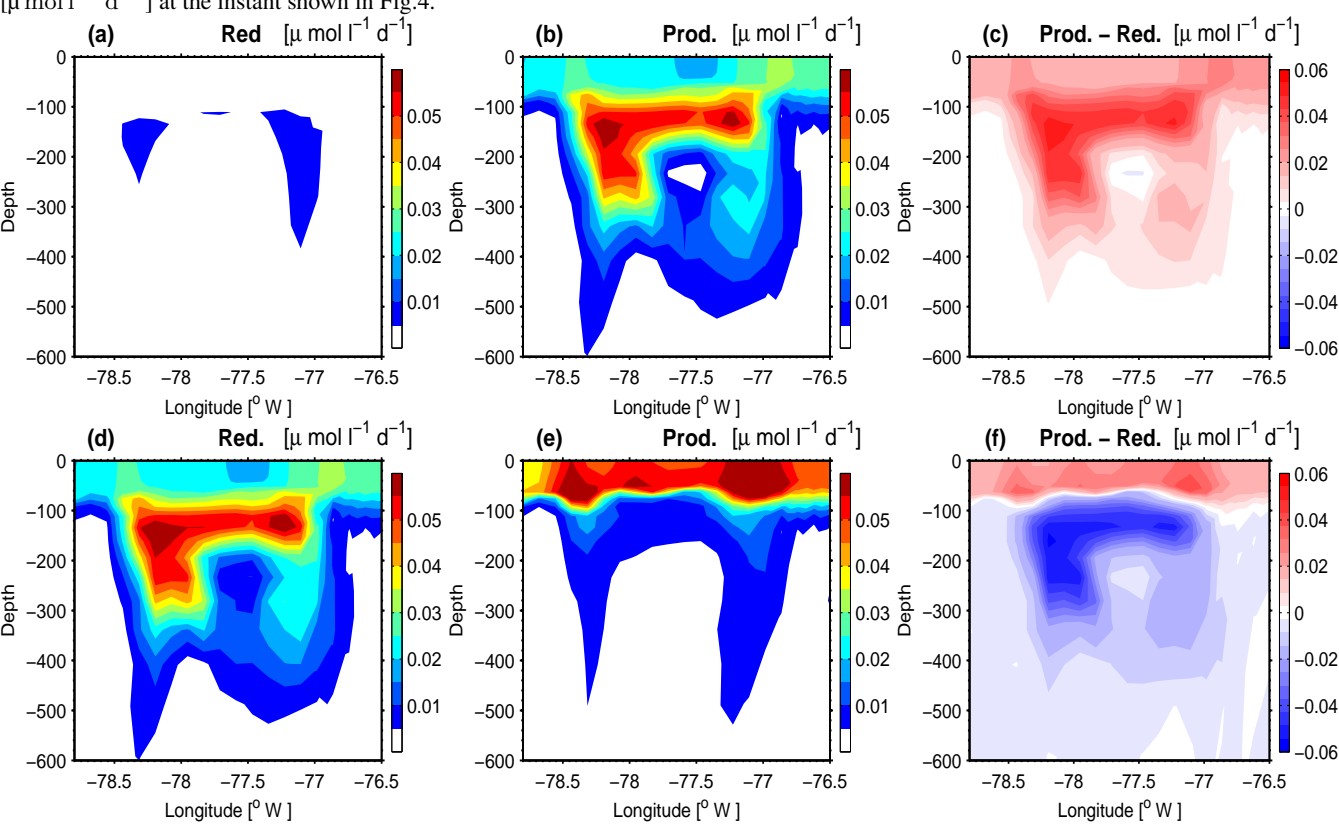

**Figure 7.** (a) Time evolution of cumulative production (full black line, [µ mol l$^{-1}$]), cumulative reduction (dashed black line, [µ mol l$^{-1}$]), production-reduction difference (blue line, [µ mol l$^{-1}$]) and available nitrate (red line, [µ mol l$^{-1}$]) within the subsurface layer of the eddy $A_{sim}$. (b) The same as (a) but for nitrite. The concentration and fluxes in (a) and (b) correspond to averaged quantities between 100-200m layer within the eddy structure. Arrow indicates the time when the sections in Figure 4 were made.

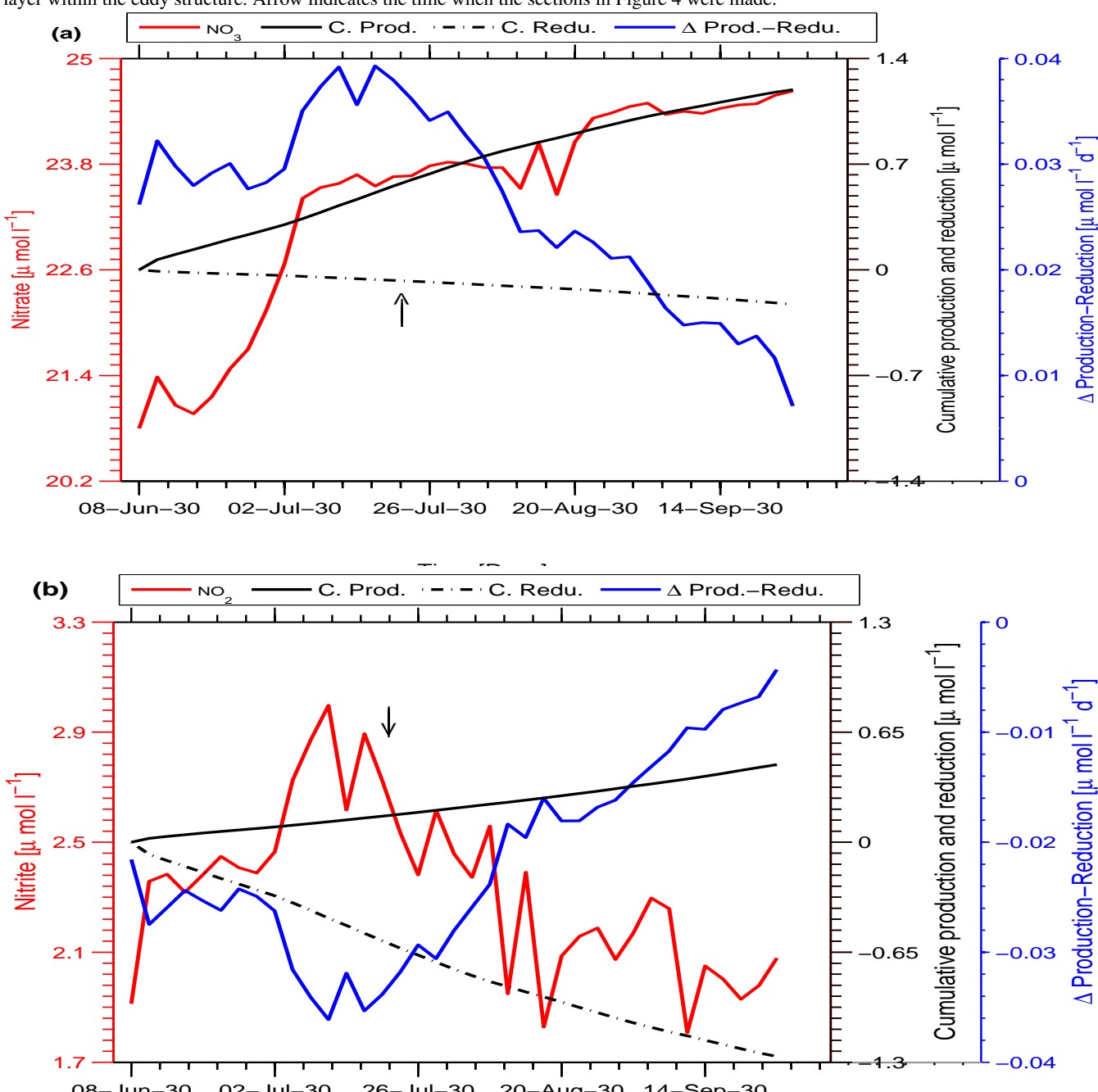

**Figure 8.** Eddy $B_{sim}$ vertical structure of: (a) nitrate reduction by denitrification [μ mol l$^{-1}$ d$^{-1}$], (b) nitrate production by nitrification [μ mol l$^{-1}$ d$^{-1}$], (c) nitrate production-reduction difference [μ mol l$^{-1}$ d$^{-1}$], (d) nitrite reduction (nitrification + denitrification + anammox, [μ mol l$^{-1}$ d$^{-1}$]), (e) nitrite production (denitrification + nitrification, [μ mol l$^{-1}$ d$^{-1}$]) and (f) nitrite production-reduction difference [μ mol l$^{-1}$ d$^{-1}$] at the instant shown in Fig.5.

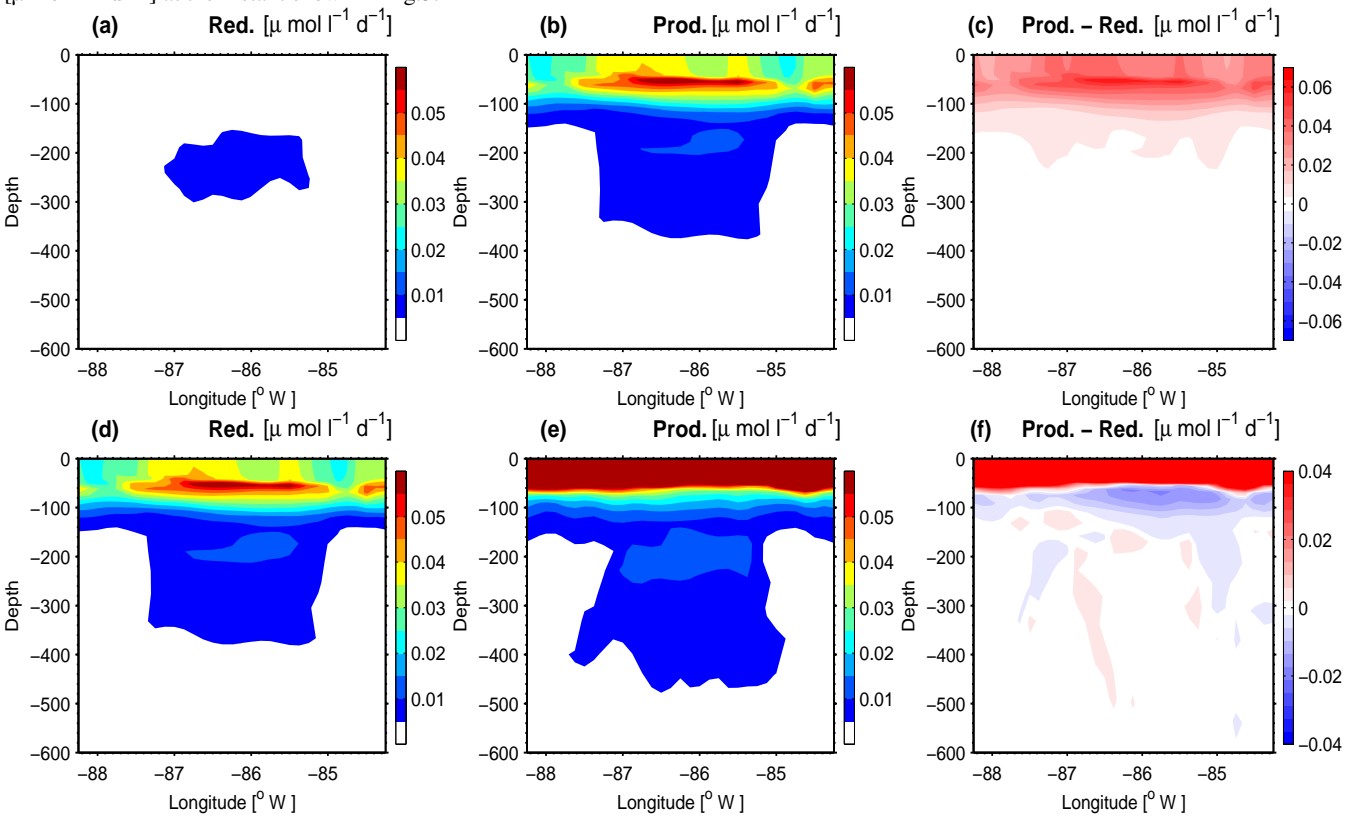

**Figure 9.** (a) Time evolution of cumulative production (full black line, $[\mu\,mol\,l^{-1}]$), cumulative reduction (dashed black line, $[\mu\,mol\,l^{-1}]$), production-reduction difference (blue line, $[\mu\,mol\,l^{-1}\,d^{-1}]$) and available nitrate (red line, $[\mu\,mol\,l^{-1}]$) within the subsurface layer of the eddy $B_{sim}$. (b) The same as (a) but for nitrite. The concentration and fluxes in (a) and (b) correspond to averaged quantities between 100-200m depth layer within the eddy structure. Arrow indicates the time when the sections in Figure 5 were made.

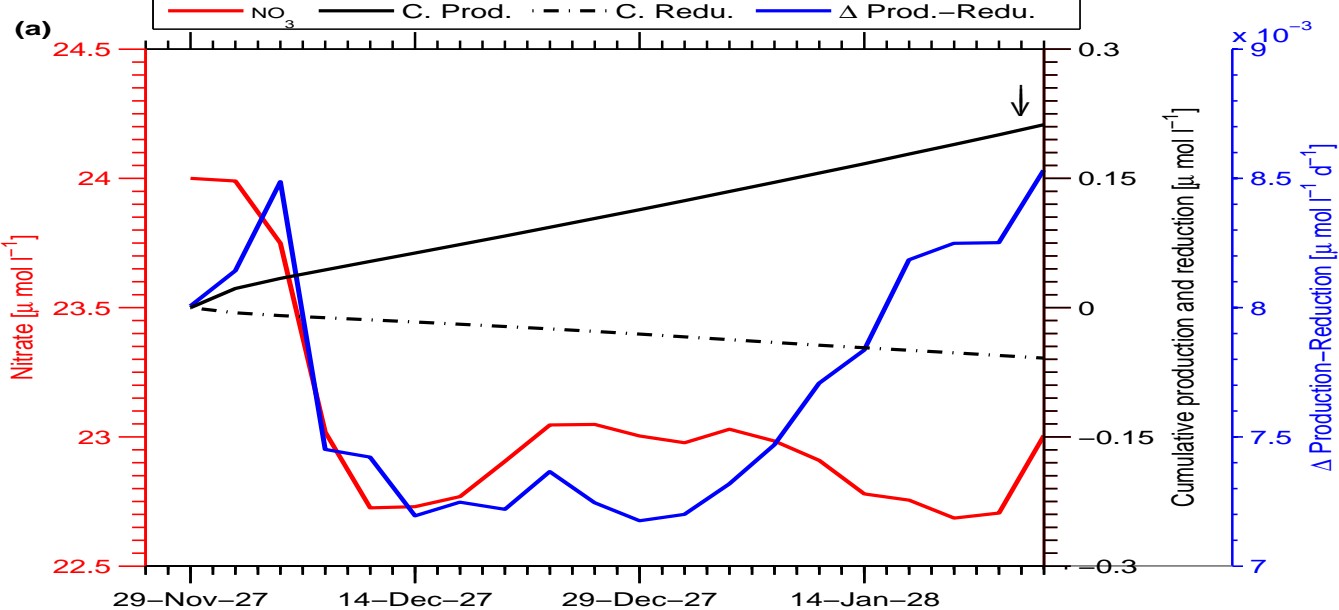

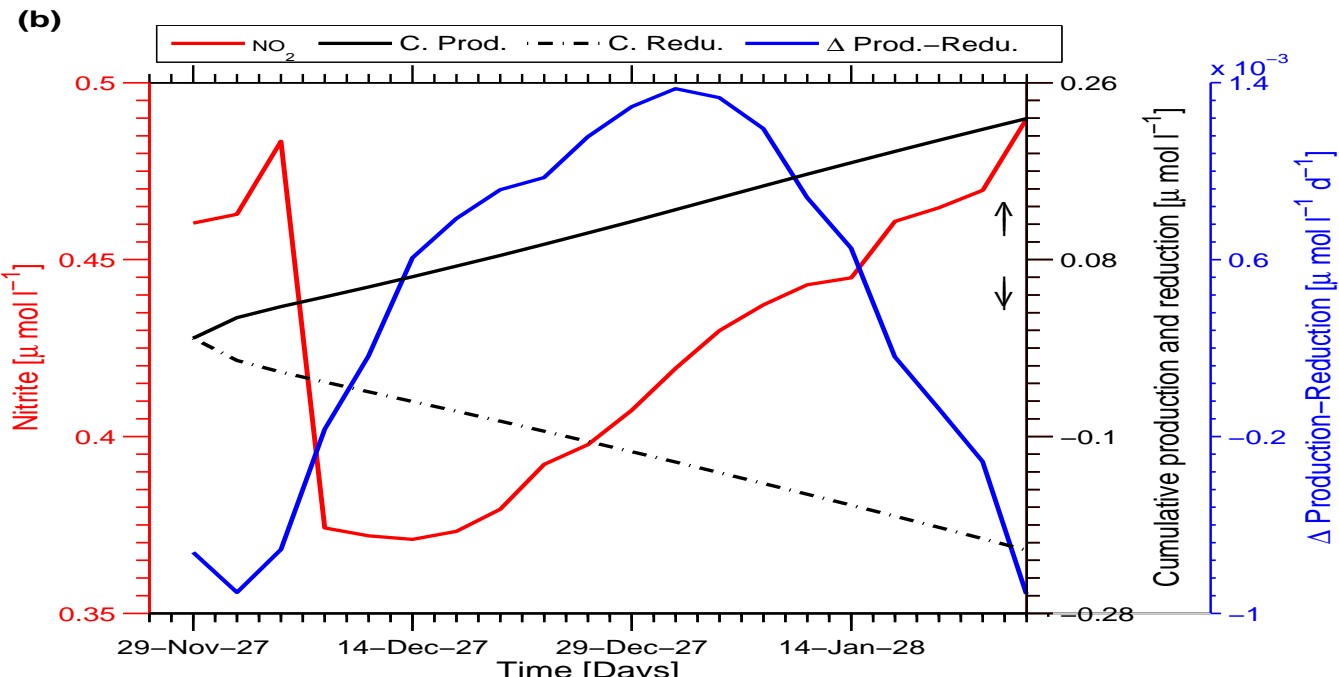

**Figure 10.** Nitrate (a) and nitrite (b) advective fluxes divergence into the eddy $A_{sim}$. Lines indicate horizontal (solid blue, $\mu\,\mathrm{mol\,l^{-1}\,d^{-1}}$]), vertical (dashed blue, $\mu\,\mathrm{mol\,l^{-1}\,d^{-1}}$]) and cumulative (black, [$\mu\,\mathrm{mol\,l^{-1}}$]) advective fluxes divergence. Red line represents the available nitrate (a) and nitrite (b) within the eddy [$\mu\,\mathrm{mol\,l^{-1}}$]. Arrows indicate the time when the sections in Figure 4 were taken.

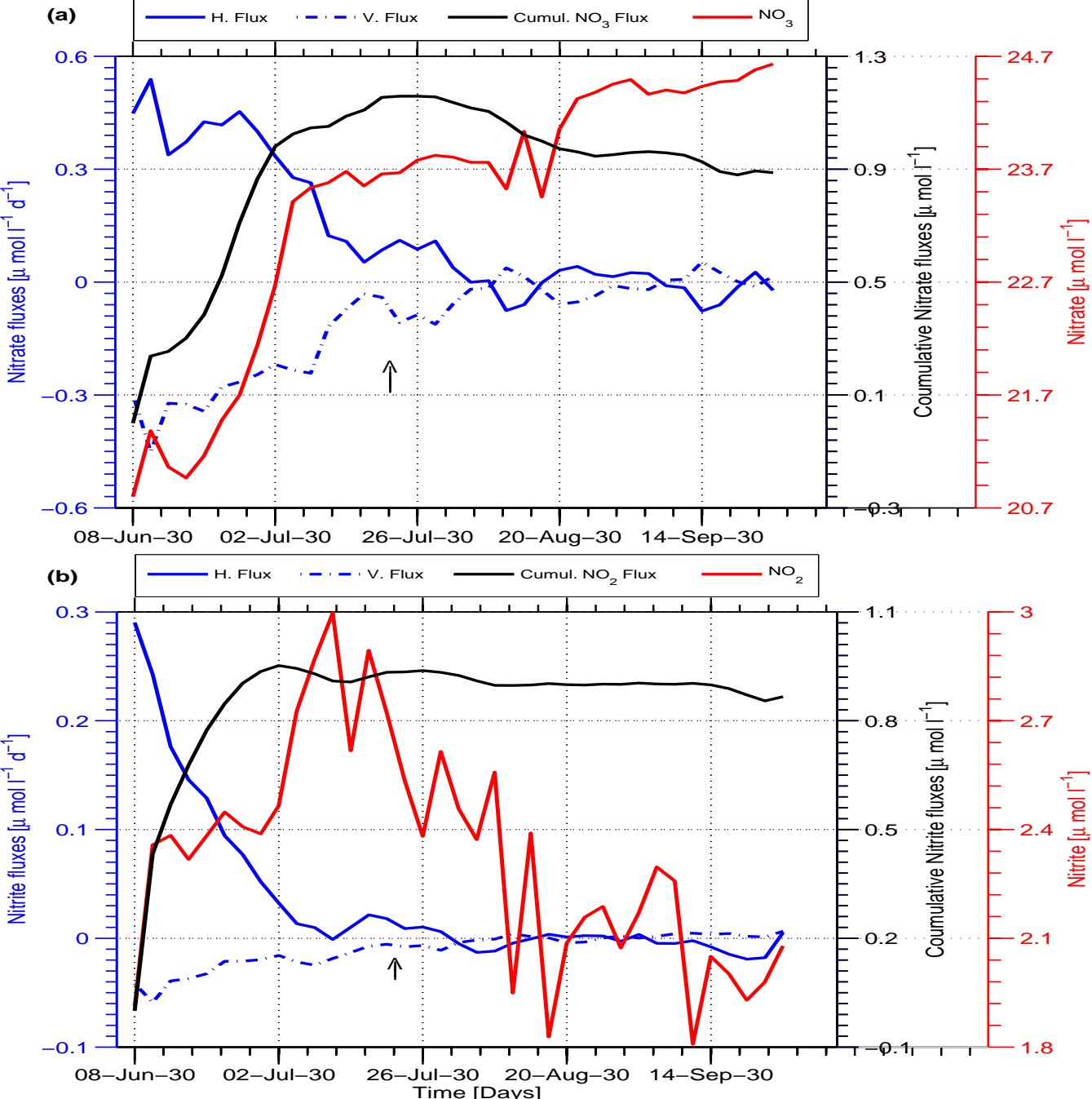

**Figure 11.** Water mass properties and nitrate distribution around the eddy A$_{sim}$. *Left panels:* Spatial distribution of simulated salinity at surface (upper panels), 100m (middle panels) and 250 m depth (bottom panels). *Middle left panels:* The same as left pannel, but for temperature. *Middle right panels:*TS relation for surface (upper panels), 100m (middle panels) and 250 m depth (bottom panels) waters. *right panels:* The same as left pannel, but for nitrate. The arrows represent the circulation patterns at the eddy's edge. Black and magenta dots in the TS diagram correspond to the water masses within the eddy at the instant after the eddy formation (magenta) and the instant shown in Fig. 4 (black dots). The magenta box in Fig 11-a indicates the region over which TS diagram is calculated.

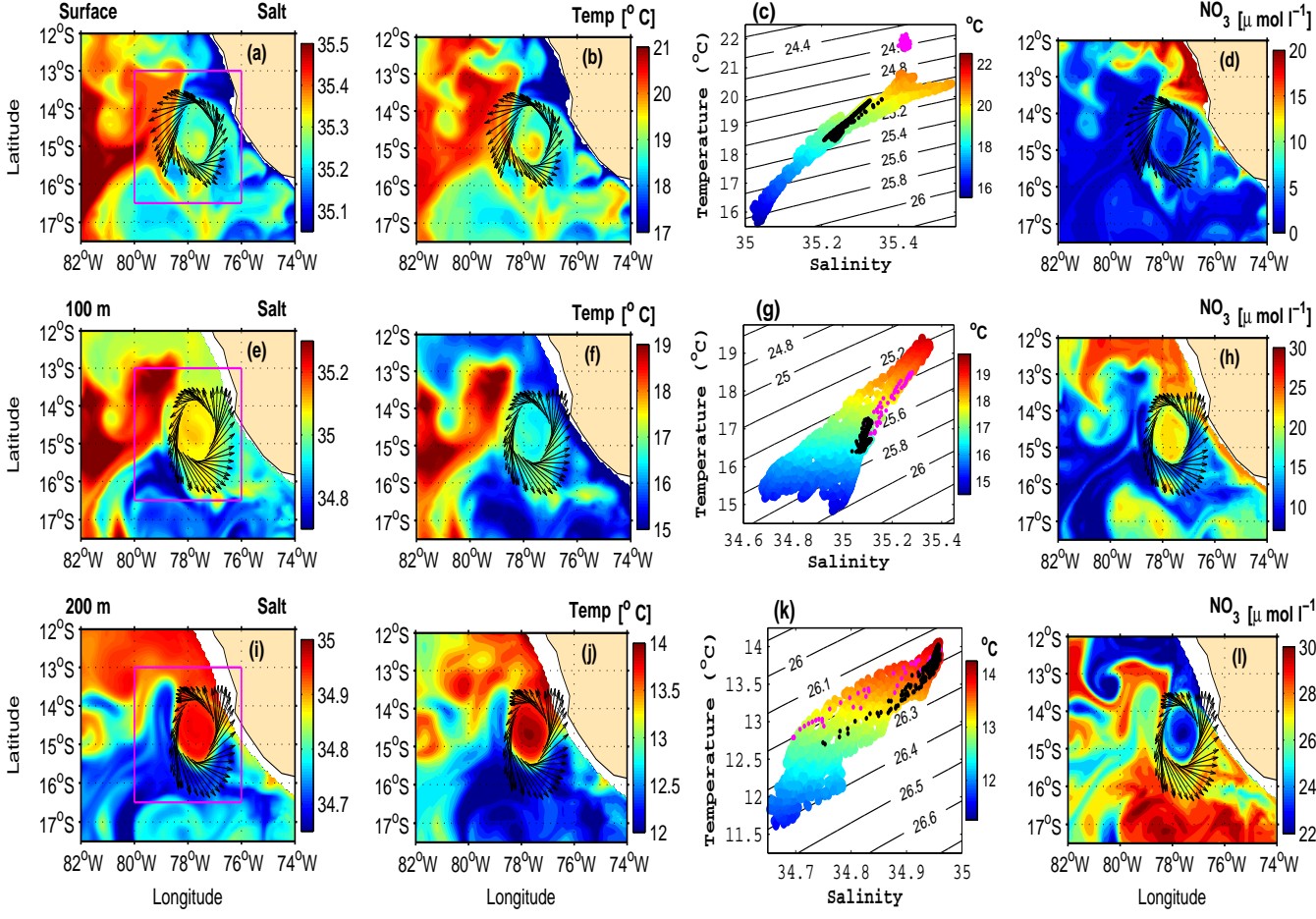

**Figure 12.** Nitrate (a) and nitrite (b) advective fluxes divergence into the eddy $B_{sim}$. Lines indicate horizontal (solid blue, $\mu\,\mathrm{mol\,l^{-1}\,d^{-1}}$]), vertical (dashed blue, $\mu\,\mathrm{mol\,l^{-1}\,d^{-1}}$]) and cumulative (black, [$\mu\,\mathrm{mol\,l^{-1}}$]) advective fluxes divergence. Red line represents the available nitrate (a) and nitrite (b) within the eddy [$\mu\,\mathrm{mol\,l^{-1}}$]. Arrows indicate the time when the sections in Figure 5 were taken.

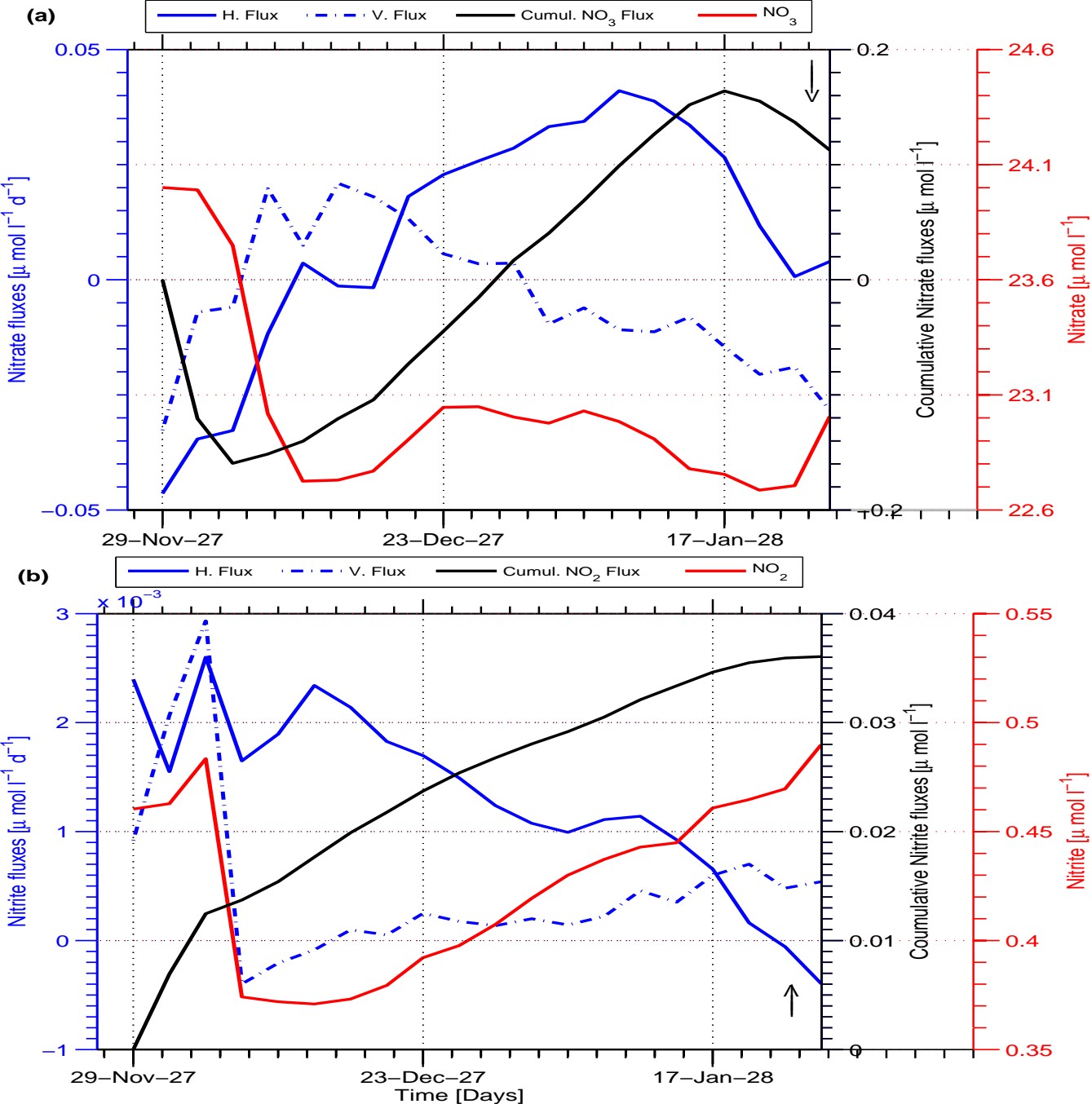

**Figure 13.** Water mass properties and nitrate distribution around the eddy B$_{sim}$. *Left panels:* Spatial distribution of simulated salinity at surface (upper panels), 100m (middle panels) and 250 m depth (bottom panels). *Middle left panels:* The same as left pannel, but for temperature. *Middle right panels:* TS relation for surface (upper panels), 100m (middle panels) and 250 m depth (bottom panels) waters. *right panels:* The same as left pannel, but for nitrate. The arrows represent the circulation patterns at the eddy's edge. Black and magenta dots in the TS diagram correspond to the water masses within the eddy at the instant after the eddy formation (magenta) and the instant shown in Fig. 4 (black dots). The magenta box in Fig 11-a indicates the region over which TS diagram is calculated.

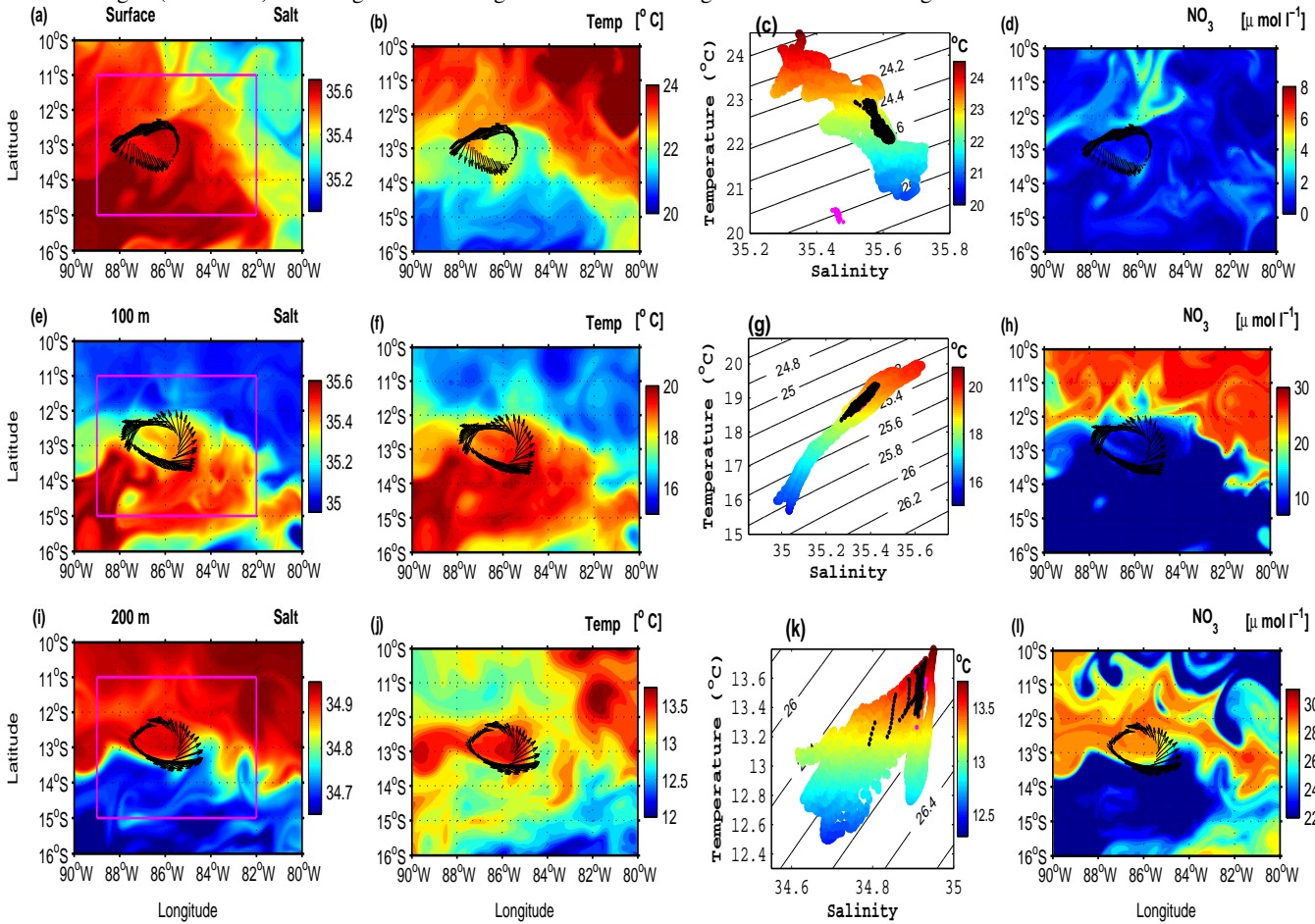

**Figure 14.** Particle distribution at the subsurface layer of anticyclonic eddies during their propagation, in summer (left) and winter (right) seasons. Anticyclonic eddies tracked at the vivinity of southern (a,b), northern (c,d) and offshore (e,f) particle-release locations. Detailed description of particle-release experiments in supplementary information. Note the change of the scale on the vertical axes.

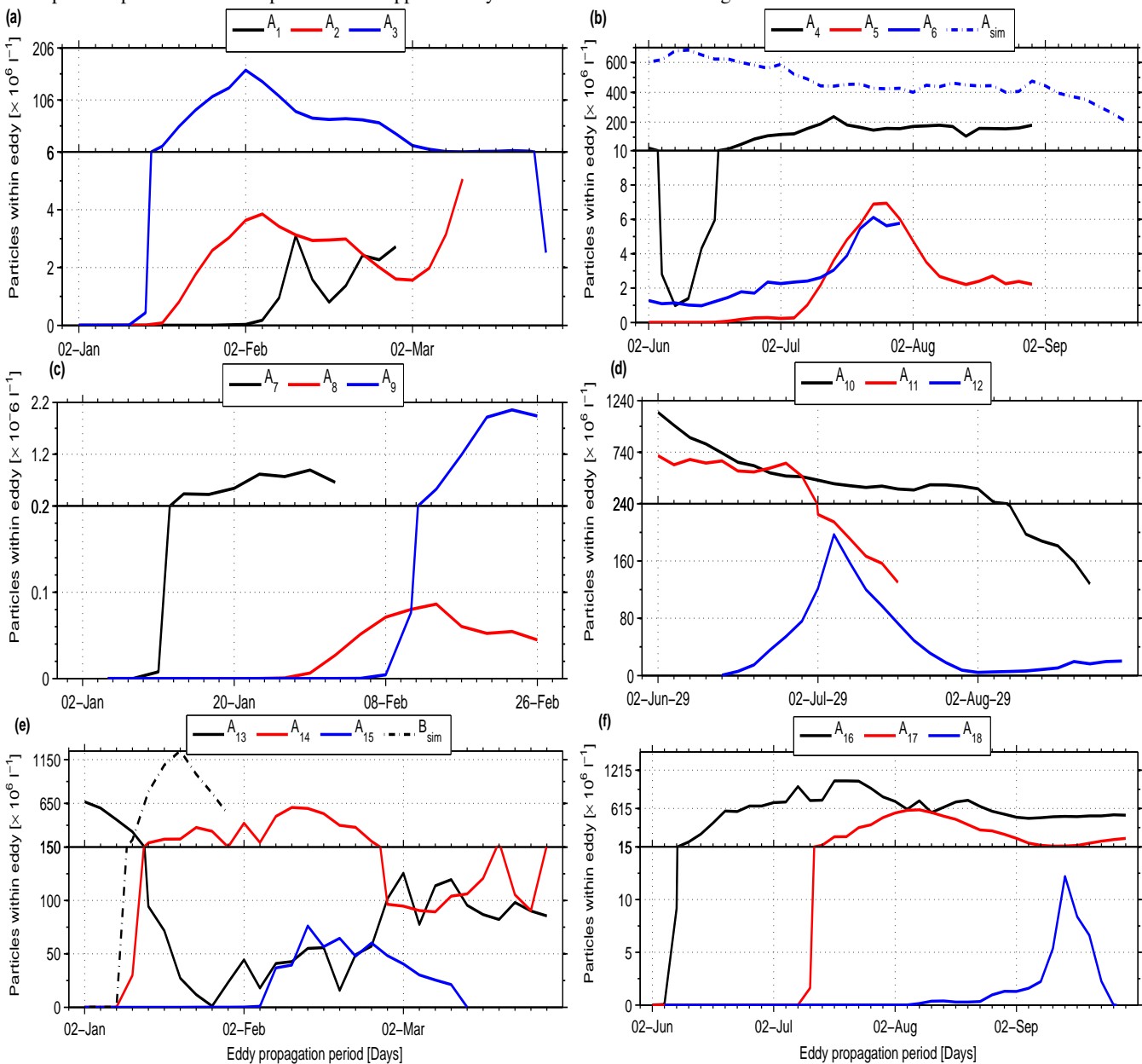