# Peer review of "Linking diverse nutrient patterns to different water masses within anticyclonic eddies in the upwelling system off Peru"

_Biogeosciences, 2016_

## Referee Comment (RC1) · Anonymous Referee #1 · 14 Jul 2016

General Comments

This paper shows results from a coupled physical and biogeochemical ocean numerical simulation. The authors identified two anticyclonic eddies off Peru in the simulation and showed that the local biogeochemical processes in the model can not explain the temporal variations in nitrate concentration within eddies. Although the authors found that local biogeochemical processes are less important than advective processes, the analyses presented by the authors were mostly focused on the role of biogeochemical processes, with some hydrographic investigations to track the origin of the advected water. Because the authors found that advective processes are important, then the advective fluxes should be analyzed directly, as they are available from the numerical

results. Another effective tools to track the origin of the watermass would be particle tracking. Without these analyses, the paper seems to lack the supporting evidences to derive the conclusions that nutrient injected into and out of the eddies by advection are playing important roles in time variations of nutrients within the eddies.

Specific Comments P1 L16 "...to enhance near-surface vertical transport and thus increase the ..." For this sentence, references should include submesoscale papers e.g. Levy et al, Mahadevan and Archer 2000.

P1 L24 "Eastern Tropical Pacific (ETSP)" Should be "Eastern Tropical South Pacific"

P3 L28 "Fig. 2-c" Should be "Fig. 3c".

P3 L29 "Fig. 3-f" Should be "Fig. 3-c".

P4 L17 "... apparently indicative of on-going denitrification within the structure, ..." As the authors have numerical results for denitrification shown in Fig. 6, these numerical indications should be mentioned here as well.

P4 L21 "... with weaker strength of the westward component ..." Why westward components are weaker? Is this result consistent with the observations?

P4 L30 "show" should be "shows"

P5 L1 "Minimum velocities" Isn't it "Maximum velocity magnitudes"? Why westward components are subsurface? Is this result consistent with the observations?

P5 L9 "Asim„" Remove one of ",".

P5 L13 "Figure 7 shows the time ..." The authors should explain how the each biogeo-chemical term, and nutrient concentration within the eddies are computed here.

P6 L25 "These water match those from ..." The authors should show if eddies often propagate northward in this region, and how PV distribute on average.

In 3.2 The authors compared the source and sink of nitrate and nitrite with the concentrations of these tracers within the eddies in Fig. 7 and 9. But cumulative production and reduction are straight lines, which means that only one rate for production and reduction at some point is used to evaluate the source and sink contributions. Is the result same if time evolutions of sink and source are taken into account? Also when the authors compute each terms within the eddies, does the averaging domain vary as the eddy evolves?

The authors should conduct more comprehensive investigations to track the origin of the water inside the eddies, by computing advective transport both along vertical and horizontal directions, and diffusive transport by parameterized turbulent diffusion. The Lagrangian particle tracking is also another effective method for this.

Also, in this paper, only two eddies are analyzed. Are these presented biogeochemical features within the eddies are representative for most of the eddies? Are cyclonic eddies not important?

---

## Referee Comment (RC2) · Anonymous Referee #2 · 1 Aug 2016

Eddies play an important role in modulating the physical and biogeochemical environments in eastern boundary upwelling systems. The authors analyze two simulated eddies in the Humbolt upwelling system. They argue that horizontal entrainment instead of biogeochemical dynamics governs biogeochemical properties inside eddies. While the mechanism is plausible, it is not sufficiently supported by the presented analysis.

Major comments:

1. Description of models is too brief. This study employs the BioEBUS model, citing a relatively recent paper [Gutknecht et al. 2013]. I don't think many readers is familiar with the model and the paper. I would suggest the authors to describe the model and the parameters in an appendix or in supplementary materials.

2. Suggest adding a figure to show the domain extent.

3. The model is insufficiently validated:

– In figs. 2(a) and (b), the model seems to capture the chlorophyll pattern correctly, but underestimate nearshore chlorophyll and overestimate offshore chlorophyll. I think this can be fixed by adjusting parameters in the biogeochemical model.

– In Figs. 2(c) and (d), the strength of the OMZ is indistinguishable due to the choice of colorbar and color scale.

– The left and middle panels of Fig. 3 suggest that the model overestimates mixed layer depth.

– In the right panel of Fig. 3, the model does not capture the high NO2 concentration in the OMZ. The authors argue that benthic process is the cause for the discrepancy. The claim is not convincing as 78.5W is quite away from any ocean bottom. I suspect the authors could adjust model parameters for oxygen-dependent nitrification/denitrification and get a better agreement.

4. In Figs. 7 and 9, averages over the upper 400 meters are presented while the difference in NO3 and NO2 between the two eddies are between 100 to 200 m. Could the authors also carry out an analysis for fluxes and concentrations between 100m and 200m?

5. While the authors claim advection is the dominant process for NO3 and NO2 concentration within eddies, there is no estimate of the advective flux. I would suggest the authors to add results for advective fluxes.

Editorial comments: 1. Fig. 3: the panels should be labeled as (a), (b), and (c) instead of (a), (c), and (e). 2. Caption for Fig. 7: "$B_{sim}$" should be "$A_{sim}$" 3. Caption for Fig. 10 needs to be revised.

---

## Referee Comment (RC3) · Anonymous Referee #3 · 8 Aug 2016

General comment : The paper focuses on the biogeochemical characteristics of mesoscale eddies in the Peru upwelling system. Due to the instabilities of the boundary currents, eddies from near the shelf break and slope trap water masses in their core and transport them offshore. Recent measurements have shown that contrasted nutrient conditions are encountered in the core of anticyclonic eddies. The goal of the paper is to investigate the nitrate and nitrite formation and evolution within two anticylonic eddies simulated by an eddy-resolving coupled dynbio model. The goal of the paper is sound and interesting as coherent eddies have an important role in the transport and mixing of properties in upwelling systems, in particular in the Peru region which hosts an intense OMZ favoring denitrification and anammox. The paper is rel-

atively well written, and the figures are of good quality (some can be improved). The simulation is carefully validated using observations, however the comparison of simulated and observed biogeochemical concentrations could be more precise, given the available observations. However, the paper has several flaws : 1) Only two eddies are studied in the model, while the model could be used to establish more robust statistics about the tmodelled eddies that are investigated. Do modelled eddies always behave as the ones that were chosen? The authors should conduct a more comprehensive eddy census with their model (what about cyclonic eddies?) 2) The two eddies that are studied are located in different regions of the domain. One is relatively close to the shelf while the other is far offshore. It was not clear whether the age of the eddies differ (I think they do), where and during which season they formed. Actually it seemed to me that the two eddies could be the same type of eddy but at different stages of its existence. 3) The discussion of the results is non-existent, and parts of the conclusion section do not reflect what has been studied in the paper. Given these remarks, I think that the paper requires a major revision before its publication.

Specific comments : P1, L17 : I do not understand the link between the processes enhancing vertical transport and basin scale effects. Please be more specific.

P1, L25 : Some references would be needed here for OMZ and denitrification/anammox

p2,L5 : Spell DNRA

p2,L27 : I do not understand the citation here.

P3,L5 : 'At the surface, the surface.. '. Rewrite.

P3,L6 : On the contrary, the EKE is reduced at the coast, which is not reproduced by the model. Why this reduction? The patterns are not really in agreement.

P3,L8 : Geostrophic currents are hard to see in this Figure as there are few isolines of sea level, so that it is impossible to see the intensified gradients associated with

the currents. POC, PCC and SEC are not identifiable, and currents do not propagate (waves do). Maybe a plot with arrows would help.

Fig1 : The vertical and horizontal black lines in panels c) and e) need to be described in the legend.

P3,L18 : A lot of other things could also impact the poleward currents: impact of the smoothed bottom topography of the model, spatial resolution of the model, resolution and temporal variability of the open boundaries, climatological run vs observations over 2008-2012, underestimated wind stress curl... I do not think you can single out one effect from the bunch at this stage.

P3,L20 : presents

P3,L24 : why is the high O2 consistent with the observed dynamics ? This sentence is unclear.

P3,L27 : in spite of => except for the deeper nitrate

P3,L28 : The observed nitrite distribution is very different from the modelled one. Also the cross-shore gradients are very different, and very difficult to see in the data. Maybe you should try to change the color scale of the observations to show a qualitative agreeement between model and observations. From this figure it is clearly not the case.

P4,L3 : 'Consistent with the patterns presented by Stramma.. ': Please elaborate the comparison with Stramma's observations. Summerize what Stramma et al. found in these eddies.

P4,L6 : 'analysed their life history. An analysis of the eddy's evolution into the future.. ': This is repetitive.

P4,L9-10 : 'the first method,.. the second..': I thought that one method was used, with two steps. Rephrase.

P4,L15 : Why not display the SSH and/or Okubo Weiss parameter, instead of meridional velocity in color scale?

P4,L17 : It would be nice to use the model to verify if indeed denitrification is on-going.

P4,L18 : I do not understand what suggests exchanges at the edge of the eddy. There is a gradient of NO3, which is expected as NO3 reduces in the eddy. Please explain what you mean here.

P4,L21 : At the time of identification by the tracking algorithm

Figs 4 and 5 : isolines for some specific O2 values would be helpful in panels b)

P4,L27 : How low ? In comparison with O2 concentration in Asim (Fig 4b) ? Please be more specific. What intermediate depths ?

P4,L28 : It is not clear from the figure that the surrounding waters are particularly well oxygenated.

P4,L30 : Please add contours in Fig 5d and 4d and be more quantitative in the text and comparison with Stramma's observations.

P4,L30 : shows

P4,L1 : The asymetric flow associated with both eddies does not no strike me as something neither very clear nor very relevant for your study. You could skip that, also in the Asim description. P5,L7 : The denomination 'identification instant' sounds a bit awkward. It should more or less correspond to the date of the eddy formation, shouldn't it ?

P5,l8 : reduction by denitrification, and production by nitrification should be in text (it is in the legend)

Figure 6 : I think it would be nice to overlay a few contours of O2 to better see the edges of the eddy.

P5,L10 : How do the production and reduction rates compare with observations ? Same remark for nitrite rates.

P5,L19 :Figures 7 and 8 are a bit puzzling: - In Figure 7a the magnitude of the nitrate changes is low in comparison to the biogeochemical trend during the first phase (8 juin-15 juillet) thus likely due to physical processes, but comparable later on. It is somewhat also true for the slow nitrite evolution after August in Fig 7b.

- I do not see a clear link between the dark blue curve (=net production) which, when cumulated in time, should be be equal to the difference between the black and the cyan linear curves. I find it surprising that the trends (black, cyan curves) are so linear. Were they computed from daily model output ?

-The quality of Figure 7 needs to be improved. Labels are deformed, it is difficult to read the dates on the x axis. Also the cummulative nitrate and nitrite consumption appear as positive values, which is misleading. How was the eddy volume defined? Which criteria was used for the subsurface?

P5,L23 : I do not see anything at the edges of the eddies in Fig 8c. Are you referring to the tiny peaks near 200-250m depth ? Is that relevant ?

Fig9ab: what happened at the end of january ? What explains these nitrate/nitrite peak ?

P6,L2 : I am not sure I clearly understand what is meant here. Do you mean eddies capture surrounding waters in their core when they are formed, and then propagate with the trapped water mass ? When the eddy has been formed, the surrounding waters are entrained on the edges of the eddy, which creates horizontal stirring.

P5,L3 : could you explain the process here ?

P6,L10 : I did not understand where Asim was formed. Is it further north or south ? The eddy is still quite close to the coast. It would be really useful to add a figure which clearly shows the trajectories of the two eddies, since you study their temporal

evolution.

Figure 10c : the colors are a bit confusing, as the y axis of the TS diagram already indicates the temperature of the WM. The magenta points in Fig 10c suggest that the eddy was formed further north, away from the area as its original WM differ from the surrounding WM.

Fig 11 : would be easier to read if there was a zoom on the eddy. No need to show the whole domain. The eddy is far from the coast.

Fig11ef : I do not see saltier water entering the eddy, but rather fresher, slightly cooler and nutrient richer water entering the eddy in Fig11e-f-h.

P6,L31 : I do not see the exchange of waters with the environment in Fig 11k. Please explain.

P7,L1-2 : These lines are not convincing. You say that there is exchange, then that it is not strong. I do not see clearly where you are going with that.

Conclusions section :

P7,L10 : I think that it is not clear why the WM are different when the eddies are formed. Is it the location ? The season ? An interesting and possibly more convincing diagnostic would be to show the nitrate/nitrite concentration in the newly formed eddies in the same Figure. Also I don't understand how you can compare these 2 eddies, which obviously have different ages. Bsim is only 2 months old according to Figure 9, and is located very far from the coast. It seems to me that it should be older than Asim which is 3.5 months old (according to Fig 7) and closer to the coast. How could the two eddies be in such different places with such age difference ? This needs clarification.

P17,L12-13 : This may be true but it remains to be demonstrated based on dedicated diagnostics. I also do not think that you can base your conclusions on the examination of only two eddies. There must be plenty of eddies in your multi-annual simulation, from which you can compute some more robust statistics.

P17,L14 : Weakened flow relative to what ? You seem to imply the WM contained in the eddy depend on the formation mechanism or site, but this is not clear.

---

## Author Comment (AC1) · 18 Oct 2016

**Detailed response to referee #2**

We are thankful to referee #2 for her/his comments and suggestions. The tuning of the biogeochemical parameter has substantially improved the simulation results presented in the now-revised version of the manuscript. In addition, the analyses of advective fluxes have allowed for a better interpretation of the processes involved in nutrient variations within the selected eddies.

Eddies play an important role in modulating the physical and biogeochemical environments in eastern boundary upwelling systems. The authors analyze two simulated eddies in the Humbolt upwelling system. They argue that horizontal entrainment instead of biogeochemical dynamics governs biogeochemical properties inside eddies. While the mechanism is plausible, it is not sufficiently supported by the presented analysis.

**Major comments:**

1. Description of models is too brief. This study employs the BioEBUS model, citing a relatively recent paper [Gutknecht et al. 2013]. I don't think many readers is familiar with the model and the paper. I would suggest the authors to describe the model and the parameters in an appendix or in supplementary materials.

Agreed. We added the respective information in a supplement.

**Addition to the supplementary information, section 1:**

**1. Biogeochemical model**

BioEBUS is a nitrogen-based model, developed from the N2P2Z2D2 by Koné et al. 2009. This model contains 12 compartments (Gutknecht et al. 2013). As described in Gutknecht et al. 2013, marine biota are represented by four compartments and comprise the first trophic level of the food web, small (nano/ picophytoplankton and microzooplankton) and large (diatoms and mesozooplankton) organisms. The phytoplankton growth is limited only by the availability of fixed nitrogen in the water column. The nitrogen cycling includes denitrification, nitrification and anammox processes, as well as uptake by phytoplankton in the sun-lit surface layer and subsequent cycling and re-cycling by the planktonic ecosystem,. The model also represents dissolved oxygen, allowing a separation of respiration processes occurring under oxic and suboxic conditions.

**1.1. Model parameters**

To simulate the biogeochemical dynamics of the ETSP, we essentially used the same parameters as in previous studies (Koné et al. 2009, Gutknecht et al. 2013, Montes et al. 2014). Some parameters are adjusted, in order to obtain a better agreement with the observed dynamics of the ETSP (Table 1). These parameters include the half saturation constant for nutrient (ammonium, nitrate and nitrite) uptake by both small and large phytoplankton classes, zooplankton (including small and large classes) feeding preferences, half saturation constant for nutrient uptake by phytoplankton and the zooplankton ingestion are adjusted to the values presented in Koné et al. 2009. The rate of first and second stage of nitrification consist of parameter values used in Gutknecht et al. 2013 and in Montes et al. 2014, respectively.

| Table 1. Adjusted parameter | used in the biogeochemical | model |
|-----------------------------|----------------------------|-------|
|-----------------------------|----------------------------|-------|

| Half saturation constant for NH4 uptake by large phytoplankton     | mmol N m -3 | 0.7  |
|--------------------------------------------------------------------|------------------------|------|
| Half saturation constant for NO2+NO3 uptake by small phytoplankton | mmol N m -3 | 1    |
| Preference of small zooplankton to small phytoplankton             | mmol N m -3 | 0.75 |
| Preference of small zooplankton to large phytoplankton             | mmol N m -3 | 0.25 |
| Preference of large zooplankton to large phytoplankton             | mmol N m -3 | 0.5  |
| Preference of large zooplankton to small zooplankton               | mmol N m -3 | 0.24 |
| Half saturation constant for ingestion by small zooplankton        | mmol N m -3 | 1    |
| Half saturation constant for ingestion by large zooplankton        | mmol N m -3 | 2    |
| Rate of first stage of nitrification                               | d-1                    | 0.9  |
| Rate of second stage of nitrification                              | d-1                    | 0.25 |

The adjustment of the phytoplankton nutrient uptake and the zooplankton dynamics constants led to a reduction of both phytoplankton and zooplankton production, and consequently export production, and an improved agreement with vertical nutrient and oxygen profiles. Adjustments of the rates of nitrification allowed a better reproduction of nitrite and nitrate distributions in our model configuration.

**2. Suggest adding a figure to show the domain extent.**

The domain extent is now added in the supplementary information.

**Addition to the supplementary information, section 2:**

**2. Model domain**

Figure SI-1 shows the extension of the model domain used in the 2 way-nesting procedure to simulate the high-resolution bio-physical dynamics of the Eastern Tropical South Pacific (ETSP). As the ETSP is strongly influenced by equatorial dynamics (Montes et al. 2010), a larger model domain with a coarser grid, covering the relevant current systems of the ETPS, is used to force the high-resolution model centered on the oxygen minimum zone off Peru.

Figure SI-1: Model bathymetry of the Eastern Tropical South Pacific. The black square denotes the zoom into the eastern tropical south Pacific oxygen minimum zone. The color denotes depth in meters. The

topography is derived from the GEBCO 1' data set.

**The model is insufficiently validated:**

- In figs. 2(a) and (b), the model seems to capture the chlorophyll pattern correctly, but underestimate nearshore chlorophyll and overestimate offshore chlorophyll. I think this can be fixed by adjusting parameters in the biogeochemical model.

Indeed, the model overestimates the offshore chlorophyll concentration. This issue, which is still recurrent in the new simulation, might be related to the model formulation. The model formulation only accounts for the phytoplankton growth limitation by nitrogen, even though growth off Peru is known to be limited by iron (Hutchins et al. 2002).

**Addition to the new version of the manuscript, section 2.1, page 3, lines 29-33:**

Although the representation of patterns of surface chlorophyll is generally good, there are biases offshore where simulated concentrations exceed the observations. We speculate that this model deficiency is related to iron limitation (c.f. Hutchins et al. 2002), which we do not explicitly account for in our current model.

**- In Figs. 2(c) and (d), the strength of the OMZ is indistinguishable due to the choice of colorbar and color scale.**

The colorbar and color scale are changed in the new version of the manuscript.

**- The left and middle panels of Fig. 3 suggest that the model overestimates mixed layer depth.**

The simulated mixed layer depth appears comparable with the observations (Fig. 1). However, in the revised version we changed some of the biogeochemical parameters (see Table 1) such that now the vertical profiles of nitrate and oxygen are more realistic (Fig. 2).

---

## Author Comment (AC2) · 24 Oct 2016

**Detailed response to the referee # 1**

We would like to thank referee 1 for the constructive comments, which have allowed a considerable improvement of the manuscript. The most substantial point the referee stated concerned the analyses of the advective fluxes of nutrients. The respective analysis has been added to the revised version of the manuscript and it further supports our conclusions.

*General Comments*

*This paper shows results from a coupled physical and biogeochemical ocean numerical simulation. The authors identified two anticyclonic eddies off Peru in the simulation and showed that the local biogeochemical processes in the model can not explain the temporal variations in nitrate concentration within eddies. Although the authors found that local biogeochemical processes are less important than advective processes, the analyses presented by the authors were mostly focused on the role of biogeochemical processes, with some hydrographic investigations to track the origin of the advected water. Because the authors found that advective processes are important, then the advective fluxes should be analyzed directly, as they are available from the numerical results. Another effective tools to track the origin of the water mass would be particle tracking. Without these analyses, the paper seems to lack the supporting evidences to derive the conclusions that nutrient injected into and out of the eddies by advection are playing important roles in time variations of nutrients within the eddies.*

We agree with Referee #1 that models offer the opportunity to perform in-depth analysis of advective processes and that this will bring more insight on the processes playing a role on nutrient variation within the eddies. Following Referee #1's advice we have computed the advective fluxes within the selected eddies. We focus on the vertical and horizontal advection because in the high-resolution configuration used here, they are of leading order (Fig.1 and Fig. 2).

[Figure]

**Figure 1.** Nitrate (upper panel) and nitrite (lower panel) advective and diffusive fluxes within the eddy $A_{sim}$. Lines indicate horizontal (black) and vertical (blue) advection [μmol l$^{-1}$d$^{-1}$], horizontal (red) and vertical diffusion [μmol l$^{-1}$d$^{-1}$]. Note the different scales on the left (advective transport) and right (diffusive transport) axis.

[Figure]

**Figure 2.** Nitrate (upper panel) and nitrite (lower panel) advective and diffusive fluxes within the eddy $B_{sim}$. Lines indicate horizontal (black) and vertical (blue) advection [μmol l$^{-1}$d$^{-1}$], horizontal (red) and vertical diffusion

[µmol l$^{-1}$d$^{-1}$]. Note the different scales on the left (advective transport) and right (diffusive transport) axis.

**Addition to the text, section 3.2, page 7 and page 8:**

…. To investigate the origin of water masses present in the selected eddies, we analyse the advective transports of both nitrate and nitrite into the eddy during the eddy's lifetime (Fig. 10 and Fig. 12). The water mass properties within the structure are also analysed and compared with the surrounding environment during different instants of the eddy's lifetime (Fig. 11 and Fig. 13). Figure 10 illustrates the nitrate and nitrite fluxes into the eddy $A_{sim.}$ It shows a strong injection of nutrients from the lateral margins of the eddy. This nutrient injection is elevated in the first months following the eddy formation. The cumulative fluxes of both nitrate and nitrite significantly increase in this period and follow the evolution of both nitrate and nitrite within the eddy. These dynamics suggest a strong exchange with the surrounding environment during this period. This is also visible in the water mass properties within the eddy structure (Fig. 11). At the surface, waters present within the eddy $A_{sim}$ are relatively cool and fresh compared to the water masses present following the eddy formation (Fig. 11-a-b)…..

The nutrient fluxes across the edge of the eddy $B_{sim}$ are presented in Figure 12. It shows a contribution of both horizontal and vertical transport to the nutrient variation within the eddy, during the eddy's lifetime. After the eddy $B_{sim}$ formation, the nitrate fluxes through the edge of the eddy $B_{sim}$ are dominantly out-going, showing a loss of nitrate to the surrounding environment (Fig. 12-a). These out-going fluxes reduce the nitrate availability within the eddy. About half a month later, the nitrate concentration within the eddy increases. This increase is to a large extent due to the nitrate supply into the eddy structure from both vertical and horizontal boundaries. On the contrary, the nitrite supply into the eddy is largest and positive in the month following the eddy formation and decreases afterwards (Fig. 12-b).

[Figure]

**Figure 10.** Nitrate (a) and nitrite (b) advective fluxes into the eddy $A_{sim}$. Lines indicate horizontal (solid blue, µmol $l^{-1}d^{-1}$), vertical (dashed blue, µmol $l^{-1}d^{-1}$) and cumulative (black, µmol $l^{-1}$) advective fluxes. Red line represents the available nitrate within the eddy [µmol $l^{-1}$]. Arrows indicate the time where the sections in Figure 4 were taken

[Figure]

**Figure 12.** Nitrate (a) and nitrite (b) advective fluxes into the eddy $B_{sim}$. Lines indicate horizontal (solid blue, $\mu$mol l$^{-1}$d$^{-1}$), vertical (dashed blue, $\mu$mol l$^{-1}$d$^{-1}$) and cumulative (black, $\mu$mol l$^{-1}$) advective fluxes. Red line represents the available nitrate within the eddy [$\mu$mol l$^{-1}$]. Arrows indicate the time where the sections in Figure 5 were taken

*Specific Comments P1 L16 ". . . to enhance near-surface vertical transport and thus increase the . . ." For this sentence, references should include submesoscale papers e.g. Levy et al, Mahadevan and Archer 2000.*

The Levy et al, 2001 and Mahadevan and Archer, 2000 have been added.

*P1 L24 "Eastern Tropical Pacific (ETSP)" Should be "Eastern Tropical South Pacific"*
Changed accordingly.

*P3 L28 "Fig. 2-c" Should be "Fig. 3C".*
Changed accordingly.

*P3 L29 "Fig. 3-f" Should be "Fig. 3-c".*
Changed accordingly.

*P4 L17 ". . . apparently indicative of on-going denitrification within the structure, . . ." As the authors have numerical results for denitrification shown in Fig. 6, these numerical indications should be mentioned here as well.*

The goal of the respective subsection is to describe the eddy dynamics as generally viewed in snapshots obtained during oceanographic measurement campaigns. Our point is that snapshots-based interpretation of the processes involved can be misleading in regions governed by mesoscale processes. The comprehensive description of the biogeochemical processes controlling the dynamics within the eddy is presented with more detail in the results and discussion section.

*P4 L21 ":. . . with weaker strength of the westward component . . ." Why westward components are weaker?*
The asymmetry in the eddy velocity is linked to the flow dynamics during the eddy generation (Chaigneau et al. 2011, Colas et al. 2012, Holte et al 2013). Anticyclonic eddies in the ETSP are generated by the instability of the subsurface poleward Peru-Chile Undercurrent (PCUC, Chaigneau et al. 2011, Colas et al. 2012). A more detailed analysis will be presented in the revised paper.

Note that, following the referee 3, the analysis of the eddy velocity asymmetry have been removed from the new version of the manuscript.

***Is this result consistent with the observations?***

The observed anticyclonic eddy in the ETSP shows also asymmetric velocities, both in the meridional and zonal velocity components (Chaigneau et al. 2011, Stramma et al. 2013, Thomsen et al. 2016).

**Addition to the text, section 2.2.1, page 4:**

Else, the eddy $A_{sim}$ presents a   subsurface velocity maximum, similar to the observed patterns within anticyclonic eddy in the ETSP (Chaigneau et al. 2011, Holte et al 2013, Stramma et al. 2013, Thomsen et al. 2016). This characteristic is related to the poleward flowing PCUC (Chaigneau et al. 2011, Colas et al. 2012, Holte et al 2013).

***P4 L30 "show" should be "shows"***

Changed accordingly.

***P5 L1 "Minimum velocities" Isn't it "Maximum velocity magnitudes"?***

It should be minimum velocity. The eastward component (positive velocity) is minimum at the surface and the westward component (negative velocity) is minimum at the subsurface. However, it is indeed preferable to describe the most prominent feature, which is the maximum velocity magnitude. This will be revised in the new version of the manuscript.

**Addition to the text, section 2.2.2, page 5:**

… Figure 5 -e and Figure 5 -f show the vertical section of the eddy $B_{sim}$ velocities.   Maximum velocities are found at the surface layers in the westward flow and at the subsurface layers in the eastward flow, respectively.

*Why westward components are subsurface? Is this result consistent with the observations?*

The subsurface velocity maxima in the eddy in the ETSP region are linked the subsurface poleward PCUC (Colas et al. 2012, Holte et al 2013). Eddies and meander type structures are shed by the Peru-Chile Undercurrent (Chaigneau et al. 2011, Colas et al. 2012, Holte et al 2013). Observations in the ETSP have found mode water anticyclonic eddies with subsurface velocity maxima, in both zonal and meridional components (Chaigneau et al. 2011, Stramma et al. 2013, Thomsen et al. 2016).

**Addition to the text, section 2.2.2, page 5:**

This circulation pattern is similar to the observed velocity within anticyclonic eddies in this region  (Chaigneau et al. 2011, Stramma et al. 2013, Thomsen et al. 2016) and likely  linked to the dynamics of the PCUC (Chaigneau et al. 2011, Colas et al. 2012, Holte et al. 2013).

*P5 L9 "Asim„" Remove one of ",".*

Changed accordingly

*P5 L13 "Figure 7 shows the time . . ." The authors should explain how the each biogeochemical term, and nutrient concentration within the eddies are computed here.*

In order to answer this comment, a subsection explaining how the biogeochemical terms as well as the advective transport are calculated is now added in the new version of the manuscript.

**Addition to the text, section 2.3, page 5:**

 **2.3 Analysis of physical and biogeochemical dynamics within the eddy**

In order to have insights into the processes controlling the nutrient distribution within the eddies, the advective transport (vertical and horizontal), sources and sinks of nutrients are analysed during the eddy evolution. The time evolution of nutrient concentrations is also presented. The eddy volume is defined here as the volume between 100m and 200m within the eddy shape. The horizontal advective transport of nutrients is calculated at the edge of the eddy structure, while the vertical transport is calculated at upper (100 m) and lower (200 m) extremes of the eddy

volume. The nutrient concentration, as well as the nutrient sources and sinks correspond to the averaged quantities within the eddy volume. The sources and sinks of nitrate consist of nitrification and denitrification, respectively. For nitrite, multiple sources and sinks are accounted. The sources are denitrification and nitrification, while the sinks consist of nitrification, denitrification and anammox.

***P6 L25 "These water match those from . . ." The authors should show if eddies often propagate northward in this region, and how PV distribute on average.***

No, eddies in the ETSP do not propagates always northward. According to Chaigneau et al. (2008) and Johnson et al. (2010), the eddies in the ETSP propagate westward and present a meridional deflection.

The eddy trajectory is now added in the new version of the manuscript and compared with the altimetry-derived eddy genesis and propagation in the region off Peru.

**Addition to the text, section 2.2.1, page 4:**

Generated in the southern part of the Peruvian shelf (around $14.5^o$ S) about 42 days before the instant presented in Figure 4, the eddy $A_{sim}$ propagates north-westward. This eddy genesis and propagation is in agreement with altimetry observations (Chaigneau et al. 2008).

**Addition to the text, section 2.2.2, page 5:**

The age of eddy $B_{sim}$ is about two months (54 days) and it was generated offshore near to $85^oW$ and $12^oS$. The place of generation of model eddy $B_{sim}$ is in agreement with the eddy genesis inferred from the altimetry observations (Chaigneau et al. 2008). Possibly detached from a meander type structure, the eddy $B_{sim}$ propagates westward and is deflected poleward.

***In 3.2 The authors compared the source and sink of nitrate and nitrite with the concentrations of these tracers within the eddies in Fig. 7 and 9. But cumulative production and reduction***

*are straight lines, which means that only one rate for production and reduction at some point is used to evaluate the source and sink contributions. Is the result same if time evolutions of sink and source are taken into account? Also when the authors compute each terms within the eddies, does the averaging domain vary as the eddy evolves?*

The cumulative production and reduction consist of the spatial averaged quantities within the eddy (now from 100-200 m depth). The full time history of the fluxes is taken into account, and the cumulative integral is smoother than the actual fluxes, but it is not a straight line. The eddy domain varies with time as the shape of the eddy is not constant. At each instant, fluxes within the eddy and the actual eddy volume are considered to compute the cumulative flux per liter eddy volume.

*The authors should conduct more comprehensive investigations to track the origin of the water inside the eddies, by computing advective transport both along vertical and horizontal directions, and diffusive transport by parameterized turbulent diffusion. The Lagrangian particle tracking is also another effective method for this.*

We thank the reviewer for this suggestion. The advective transports are now included in the new version of the manuscript (section 3.2, page 7 and page 8).

*Also, in this paper, only two eddies are analyzed. Are these presented biogeochemical features within the eddies are representative for most of the eddies? Are cyclonic eddies not important?*

Indeed, in the paper only two anticyclonic eddies (mode water eddies) are analysed. According to Colas et al. (2012), the mode water anticyclonic eddies (also known as subsurface anticyclonic vortices) are the predominant anticyclonic eddy feature in this region. The main goal of this paper is to understand the diverse nutrient patterns observed at the subsurface layer of two mode-water anticyclonic eddies off Peru.

The cyclonic eddies also play an important role on the nutrient dynamics in this region. The uplifted isolines within the cyclonic eddies (McGillicuddy et al. 1998) may bring upward the deep nutrient-rich waters, contributing to the nutrient cycle in the upper layers. Note, however,

that this is beyond the scope of the manuscript.

Nonetheless, in order to gain more insight into the processes governing the nutrient dynamics within anticyclonic eddies, we conducted a particle-release experiments. Anticyclonic eddies tracked at the vicinity of the particle-release locations have trapped particles within their centre. The number of particles within these eddies varied in time, suggesting an exchange with surrounding environment during their propagation. This result supports the hypothesis that nutrient patterns within anticyclonic eddies can be strongly affected by physical exchange processes with surrounding waters.

**Addition to the text, section 2.4, page 6:**
**2.4 Particle release experiment**

In order to have a more general overview of the processes controlling the dynamics of the eddies in the ETSP, we conduct a particle-release experiments and analyse the anticyclonic eddies that are in the vicinity of the particle-release locations at the time of the release. In these experiments, particles are released in three different locations in ETSP: (1) along the shelf between $13^o$S - $15^o$S, (2) along the shelf between $9^o$S $11^o$S, and (3) offshore between $13^o$S-$15^o$S in latitude and $85^o$W-$86^o$W (cf. Fig. SI-2 in supplementary information). The particles are released in the entire water column on the shelf and in the upper 300 m at the offshore site, in early austral summer (January) and early winter (June) of the last three climatological years of the model simulation.

**Addition to the text, section 3, page 8 and page 9:**
**3.3 Eddy stirring and nutrient entrainment**

The eddies $A_{sim}$ and $B_{sim}$ showcase that the nutrient supply by physical dynamics is the dominant mechanism that controls simulated (diverse) nutrient pattern within the eddies. The nutrient exchange with surrounding waters occurs throughout the entire lifetime of the eddies.. This indicates that the nutrient availability in the vicinity of the eddy plays a role for the nutrient distribution within the eddy's structure. To elucidate this suggestion, we carried out particlerelease experiments (subsection 2.4) and analysed the eddies that passed and/or were generated close to the particle-released areas. Figure 14 illustrates the particle distribution in the subsurface layer (between 100 - 200m depth) of anticyclonic eddies interior during their propagation. From the early stages on, particles are entrained and trapped within the eddy structures.

[Figure]

**Figure 14.** Particle distribution at the subsurface layer of anticyclonic eddies during their propagation, during summer (left) and winter (right) seasons. Anticyclonic eddies tracked at the vicinity of southern (a,b), northern (c,d) and offshore (e,f) particle-release locations. Detailed description of particle-release experiments can be found in the supplementary information.

These particles are transported offshore during propagation of the eddies. Every tracked eddy shows a pronounced temporal variation of the amount of particles within the structure, an indicative of exchange of properties with surrounding waters. This behavior occurs in eddies tracked during both austral summer (Fig. 14-a,c,e ) and austral winter (Fig. 14-b,d,f ).

**Addition to the text, section 4, page 10:**

Anticyclonic eddies tracked during  the particle-release experiments corroborate this suggestion and show the occurrence of water mass exchange between the eddy and the surrounding environment. Particle numbers within these eddies are repeatedly increased and decreased, showing a loss and gain of quantities to/from the surrounding environment.

**Addition to the text, section 5, page 10:**

In a more general context the particle-release experiments realized in this study also emphasize the role of water mass exchange between eddies and the surrounding environment for the temporal evolution of properties within the eddy structure.

**Addition to the supplementary information, section 3, page 2:**

 **3. Particle-release experiments**

In Figure SI-2 are illustrated the locations of particle release (light blue) as well as the tracked anticyclonic eddies (filled circles and triangles) during the model particle-release  experiments. The particle-release experiments, which consisted of releasing inactive Lagrangian particles along the shelf and off Peru, are conducted to investigate the capability of anticyclonic eddies to exchange water masses with the surrounding environment. In order to cover possible seasonality effects, the particles were released in both summer and winter seasons of the southern hemisphere.

[Figure]

**Figure SI-2.** Particle release sites (light blue) and the trajectory of tracked anticyclonic eddies (filled circles and triangles) during the particle release experiment. Colour in tracked eddies correspond to model years, with black for year 28, red for year 29 and blue for year 30. Eddies tracked during summer and winter are represented in filled circles and triangles respectively.

**Addition to the text, section 5, page 10 and page 11:**

This physical exchange of water mass properties with the ambient environment is likely to contribute to shaping the nutrient patterns within cyclonic eddies.

**References**

Chaigneau et al. (2008). *Mesoscale eddies off Peru in altimeter records: Identification*

*algorithms and eddy spatio-temporal patterns*, Progress in Oceanography, 79, doi:10.1016/j.pocean.2008.10.013, 2008.

Chaigneau et al. (2011). *Vertical structure of mesoscale eddies in the eastern South Pacific Ocean: A composite analysis from altimetry and Argo profiling floats*, Journal of Geophysical Research, 116, doi:10.1029/2011JC007134, 2011.

Colas ,et al. (2012), *Heat balance and eddies in the Peru-Chile current system*, Climate Dynamics, 39, 509-529, 2012.

Holte et al. (2013). Structure and surface properties of eddies in the southeast Pacific Ocean. Journal of Geophysical Research Oceans, 118, doi:10.1002/jgrc.20175.

Johnson et al. (2010), *Equatorial Pacific 13°C water eddies in the eastern subtropical South Pacific Ocean,* Journal Physical Oceanography, 40, doi: 10.1175/2009JPO4287.1

McGillicuddy et al. (1998). *Influence of mesoscale eddies on new production in the Sargasso Sea*. Nature, 394, 263-266.

Stramma et al. (2013), *On the role of mesoscale eddies for the biological productivity and biogeochemistry in the eastern tropical Pacific Ocean off Peru*, Biogeosciences, 10, 7293-7306, doi: 10.5194/bg-10-7293-2013, 2013.

Thomsen et al. (2016). *The formation of a subsurface anticyclonic eddy in the Peru-Chile Undercurrent and its impact on the near-coastal salinity, oxygen, and nutrient distributions*, Journal of Geophysical Research Oceans, 120, doi: 10.1002/2015JC010878, 2015.

---

## Author Comment (AC3) · 24 Oct 2016

*Detailed response to referee # 3*

We would like to thank referee # 3 for the constructive comments, which have allowed a considerable improvement of the manuscript. Detailed responses to the referee #3's concerns are addressed below. Respective changes are highlighted in bold in the accordingly-revised manuscript.

*General comment :*

*The paper focuses on the biogeochemical characteristics of mesoscale eddies in the Peru upwelling system. Due to the instabilities of the boundary currents, eddies from near the shelf break and slope trap water masses in their core and transport them offshore. Recent measurements have shown that contrasted nutrient conditions are encountered in the core of anticyclonic eddies. The goal of the paper is to investigate the nitrate and nitrite formation and evolution within two anticyclonic eddies simulated by an eddy-resolving coupled dynbio model. The goal of the paper is sound and interesting as coherent eddies have an important role in the transport and mixing of properties in upwelling systems, in particular in the Peru region which hosts an intense OMZ favoring denitrification and anammox. The paper is relatively well written, and the figures are of good quality (some can be improved). The simulation is carefully validated using observations, however the comparison of simulated and observed biogeochemical concentrations could be more precise, given the available observations. However, the paper has several flaws :*

*1) Only two eddies are studied in the model, while the model could be used to establish more robust statistics about the modelled eddies that are investigated.*

We agree with referee 3 that the model could be used to establish a more robust statistic evaluation of eddy dynamics and their role in the nitrogen cycle. But this is not the main goal of this study. The present study aims to understand the diverse nutrient patterns observed within two anticyclonic eddies off Peru (Altabet et al., 2012; Stramma et al.,

2013). Investigating two simulated anticyclonic eddies with nutrient patterns similar to the observations, the aim was to identify respective processes that cause these patterns in our model. To do so we analysed the life history of the simulated eddies and explored both local and remote processes that affect the nutrient distribution within the eddies.

***Do modelled eddies always behave as the ones that were chosen? The authors should conduct a more comprehensive eddy census with their model (what about cyclonic eddies?)***

The main goal of the present study is to understand the processes responsible for the diverse nutrient patterns observed at the subsurface layer of two mode-water anticyclonic eddies off Peru (Stramma et. al. 2013). For that propose, we identified in the model simulation two eddies with similar nutrient dynamics as the observations and analysed both physical and biogeochemical dynamics through their lifetime.

As concerns cyclonic, they do play a role on nutrient cycle of the eastern tropical south Pacific. Because of the vertical displacement of the isolines within the cyclonic eddies (McGillicuddy et al. 1998), cyclonic eddies might increase the transport of nutrients to the sun-lit surface.

A more comprehensive eddy census as well as the dynamics within cyclonic eddies are beyond the scope of the manuscript.

***2) The two eddies that are studied are located in different regions of the domain. One is relatively close to the shelf while the other is far offshore. It was not clear whether the age of the eddies differ (I think they do), where and during which season they formed. Actually it seemed to me that the two eddies could be the same type of eddy but at different stages of its existence.***

Indeed, the two selected eddies are not located in the same region. The selection of the eddies was primarily guided by the observed eddies off Peru. In Stramma et. al. (2013), the

low-nitrate and high-nitrite eddy was observed close to the shelf, where the eddies are formed by the instability of the poleward undercurrent. The high-nitrate and nitrite eddy was observed offshore.

The age of the selected eddies at the instant presented in Figure 4 and Figure 5, respectively, is different. However, the difference is relatively small. The eddy $A_{sim}$ is 42 days old while $B_{sim}$ is 54 days old.

The trajectories, the origin of the eddies as well as their ages are now added in the new version of the manuscript.

**Addition to the text, section 2.2.1, page 4:**

Generated in the southern part of the Peruvian shelf (around 14.5$^o$ S) about 42 days before the instant presented in Figure 4, the eddy $A_{sim}$ propagates north-westward. This eddy genesis and propagation is in agreement with altimetry observations (Chaigneau et al. 2008).

**Addition to the text, section 2.2.2, 5:**

The age of eddy $B_{sim}$ is about two months (54 days) and it was generated offshore near to 85$^o$W and 12$^o$S. The place of generation of model eddy $B_{sim}$ is in agreement with the eddy genesis inferred from the altimetry observations (Chaigneau et al. 2008). Possibly detached from a meander type structure, the eddy $B_{sim}$ propagates westward and is deflected poleward.

*3) The discussion of the results is non-existent, and parts of the conclusion section do not reflect what has been studied in the paper. Given these remarks, I think that the paper requires a major revision before its publication.*

A discussion section is now added in the new manuscript. The conclusion section is rewritten in the new version. We hope the new structure and organization of the material is clearer and easier to follow for the reader.

**Addition to the text, section 4, page 8, page 9 and page 10:**

[revised manuscript text omitted]

*Specific comments :*

*P1, L17 : I do not understand the link between the processes enhancing vertical transport and basin scale effects. Please be more specific.*

According to Oschlies (2002), eddy-induced vertical and horizontal nutrient supply into the euphotic zone increases the biological production along the margins of the North Atlantic oligotrophic gyre. A later study by Eden and Dietze (2009) found no significant relation between changes in eddy activity with biological production.

This is now removed in the new version of the manuscript.

*P1, L25 : Some references would be needed here for OMZ and denitrification/ anammox*

Changed accordingly.

**Addition to the text, section 1, page 1 and page 2:**

The region is known for oxygen-deprived waters at intermediate depth (Chavez et al. 2008) that host anoxic biogeochemical cycling of organic matter such as denitrification (Codispoti and Christensen, 1985, Farias et al. 2009) and anaerobic ammonium oxidation (anammox) (Hamersley et al. 2007, Lam et al. 2009).

*P2,L5 : Spell DNRA*

Changed accordingly.

**Addition to the text, section 1, page 2:**

... local denitrification and/or dissimilatory nitrate reduction to ammonia (DNRA).

*P2,L27 : I do not understand the citation here.*

Sentence rewritten.

**Addition to the text, section 1, page 2:**

According to the authors, the nitrate deficit in the low-nitrate eddy is caused by local denitrification and/or dissimilatory nitrate reduction to ammonia (DNRA)

*P3,L5 : 'At the surface, the surface.. '. Rewrite.*

Changed accordingly.

**Addition to the text, section 1 :**

At the surface, the eddy kinetic energy (EKE) is increased …

***P3,L6 : On the contrary, the EKE is reduced at the coast, which is not reproduced by the model. Why this reduction? The patterns are not really in agreement.***

I agree with referee 3 that the model does not meticulously reproduce the observed EKE. There is a stronger EKE in the inner shelf south of $17^o$S in the model simulation, which is not in the observations. Many factors may contribute to this discrepancy: The proximity to the coast may induce observational errors (Strub, 2001, Saraceno et al. 2008), there may be model bias. Our point here is that the general patterns are reproduced by the model.

**Addition to the text, section 2.1, page 3:**

Despite the fact that the model captures the general patterns of surface EKE, some differences are notable between the simulated and observed dynamics. These discrepancies might be related to biases both in the AVISO data (due to the proximity to the coast, Strub, 2001, Saraceno et al. 2008), and in the model solution.

***P3,L8 : Geostrophic currents are hard to see in this Figure as there are few isolines of sea level, so that it is impossible to see the intensified gradients associated with the currents. POC, PCC and SEC are not identifiable, and currents do not propagate (waves do). Maybe a plot with arrows would help.***

The sea surface height isolines are enhanced in the revised Figure 1 and the current names are also marked in the Figure. The sentence is now rewritten.

**Addition to the text, section 2.1, page 3:**

 The surface currents, namely the Peru Oceanic Current (POC) in the open ocean and the Peru Coastal Current (PCC), transport southern-origin waters north-westward, contributing to the westward South Equatorial Current…

*Fig1 : The vertical and horizontal black lines in panels c) and e) need to be described in the legend.*

Changed accordingly.

**Addition to the text, in legend of Figure 2 :**
The black lines in (c, e) are the upper and onshore limits of the observed velocities.

*P3,L18 : A lot of other things could also impact the poleward currents: impact of the smoothed bottom topography of the model, spatial resolution of the model, resolution and temporal variability of the open boundaries, climatological run vs observations over 2008-2012, underestimated wind stress curl... I do not think you can single out one effect from the bunch at this stage.*

Sentence rewritten.

**Addition to the text, section 2.1, page 3:**
There are many factors that might have contributed to this discrepancy, among them the low resolution of the boundary conditions used in this simulation, smoothed bottom topography of the model.

*P3,L20 : presents*
Changed accordingly.

*P3,L24 : why is the high O2 consistent with the observed dynamics ? This sentence is unclear.*

Sentence rewritten.

**Addition to the text, section 2.1, page 3:**

The simulated oxygen content at 300 m depth is relatively high around the equator, as shown in the observed dynamics.

*P3,L27 : in spite of => except for the deeper nitrate*

Changed accordingly.

*P3,L28 : The observed nitrite distribution is very different from the modelled one. Also the cross-shore gradients are very different, and very difficult to see in the data. Maybe you should try to change the color scale of the observations to show a qualitative agreement between model and observations. From this figure it is clearly not the case.*

Indeed, the simulated nitrite is different from the observations. However, the model captured the general patterns observed in this region. Also, following the comments of referee 2, a new simulation with adjusted parameters was performed and now shows a better agreement with the observations (Figure 1).

[Figure]

**Figure 1:** Vertical section of nitrite [μ mol /l] concentrations along 12oS. Simulated results correspond to climatological December. The observed nitrate distribution is based on measurements from the cruise M91, December 2012.

*P4,L3 : 'Consistent with the patterns presented by Stramma.. ': Please elaborate the comparison with Stramma's observations. Summerize what Stramma et al. found in these eddies.*

Changed accordingly.

**Addition to the text, section 2.2, page 4:**

In this study, Stramma et al. (2013) have found two anticyclonic mode water eddies featuring different nutrient patterns at the subsurface layer. One eddy, which was located on the shelf, presented high nitrate and low nitrite in its centre. The second, long lived offshore eddy, exhibited high nitrate and high nitrite concentration in its centre.

*P4,L6 : 'analysed their life history. An analysis of the eddy's evolution into the future..': This is repetitive.*

Sentence rewritten.

**Addition to the text, section 2.2, page 4:**

.... we tracked the eddies on time and analysed both physical and biogeochemical dynamics within the eddy structure.

*P4,L9-10 : 'the first method,.. the second..': I thought that one method was used, with two steps. Rephrase.*

Sentence rewritten.

**Addition to the text, section 2.2, page 4:**

The eddy shape is denoted by the largest connected area inside a closed contour of SSH (Chelton et al., 2011) where vorticity dominates strain (i.e where the Okubo–Weiss parameter is negative, Chelton et al., 2007).

*P4,L15 : Why not display the SSH and/or Okubo Weiss parameter, instead of meridional velocity in color scale?*

Changed accordingly. The SSH is now displayed in Figure 4a and Figure 5a.

*P4,L17 : It would be nice to use the model to verify if indeed denitrification is on-going.*
The authors agree with the above comment. The present work was carried out to understand if the low nitrate observed within an anticyclonic eddy off Peru was related to an on-going denitrification. By showing snapshots only (Figure 4 and Figure 5) this can not be achieved.

There is on-going denitrification in the model. This feature and other processes controlling the nitrate (and nitrite) distribution are now analysed in more detail in the results section.

*P4,L18 : I do not understand what suggests exchanges at the edge of the eddy. There is a gradient of NO3, which is expected as NO3 reduces in the eddy. Please explain what you mean here.*

With this sentence we meant that the water masses at the edge of the eddy can be from the surrounding environment, entrained by the eddy stirring. Our computation of the advective fluxes (new Figure 10 and Figure 12) shows that this process can be significant for the nutrient evolution within the eddies.

This is now clarified.

**Addition to the text, section 2.2.1, page 4:**
This high nitrate at the edge might be entrained from the surrounding environment by horizontal stirring.

*P4,L21 : At the time of identification by the tracking algorithm*
*Figs 4 and 5 : isolines for some specific O2 values would be helpful in panels b)*

Changed accordingly.

*P4,L27 : How low ? In comparison with O2 concentration in Asim (Fig 4b) ? Please be more specific. What intermediate depths ?*

Sentence rewritten.

**Addition to the text, section 2.2.1:**

**page 4:**

This eddy presents oxygen-depleted intermediate waters (between 100-400 m depth) in its centre, …

**page 5:**

This eddy also presents extremely low oxygen concentrations at intermediate depths in its center, which is similar to eddy $A_{sim}$ .

*P4,L28 : It is not clear from the figure that the surrounding waters are particularly well oxygenated.*

Sentence is removed from the new version of the manuscript.

*P4,L30 : Please add contours in Fig 5d and 4d and be more quantitative in the text and comparison with Stramma's observations.*

Changed accordingly.

**Addition to the text, section 2.2.2, page 5:**

Despite reproducing the subsurface nitrite maxima, the simulated concentrations are an order of magnitude lower compared to the concentration within the observed eddy.

***P4,L30 : shows***

Changed accordingly.

***P4,L1 : The asymetric flow associated with both eddies does not no strike me as something neither very clear nor very relevant for your study. You could skip that, also in the Asim description. P5,L7 : The denomination 'identification instant' sounds a bit awkward. It should more or less correspond to the date of the eddy formation, shouldn't it ?***

The asymmetric part in the eddy description is removed.

The identification instant corresponds to the time of Figure 4 and Figure 5. This does not correspond to the date of the eddy formation. For the eddy $A_{sim}$, this instant correspond to around 42 days after the eddy formation. The identification instant of the eddy $B_{sim}$ (the open ocean eddy), sampling period is around 52 days after the eddy formation.

***P5,l8 : reduction by denitrification, and production by nitrification should be in text (it is in the legend)***

Changed accordingly.

***Figure 6 : I think it would be nice to overlay a few contours of O2 to better see the edges of the eddy.***

Changed accordingly.

***P5,L10 : How do the production and reduction rates compare with observations ? Same remark for nitrite rates.***

The production and reduction rates of nitrate and nitrite are comparable with the rates observed in the ETSP (Ward et al., 2009, Kalvelage et al. 2013).

**Addition to the text, section 3.1, page 6:**
Note that the magnitude of simulated nutrient production and reduction rates, which varies from 0.01 umol/l/d and 0.1 umol/l/d (Fig. 7 to Fig. 10), are in the range of the observed denitrification, nitrification and anammox rates in the upwelling system off Peru (Ward et al., 2009, Kalvelage  et al. 2013).

*P5,L19 :Figures 7 and 8 are a bit puzzling: - In Figure 7a the magnitude of the nitrate changes is low in comparison to the biogeochemical trend during the first phase (8 juin-15 juillet) thus likely due to physical processes, but comparable later on. It is somewhat also true for the slow nitrite evolution after August in Fig 7b.*

We are sorry for this confusion. Maybe the colour code in the Figure 7 was not explicit enough, because the comment is apparently not in agreement with the results displayed in this Figure. In Figure 7a, the nitrate concentration increases by about 1mol/l from 8 June-15 July. This increase is way larger than changes that could be related to biogeochemical sources and sinks. If we take the net production, for instance, the net change are about 0.01mol/l/d in the same period, only.

Indeed, there is a slow evolution of nitrite concentration after August as the loss by biogeochemical processes is also slowly evolving.

The contribution of biogeochemical processes   to nutrient availability within the eddy appears to be  extremely low compared to changes in nutrient concentration.

We did increase the precision in the new manuscript to make it clearer.

*- I do not see a clear link between the dark blue curve (=net production) which, when cumulated in time, should be be equal to the difference between the black and the cyan*

*linear curves. I find it surprising that the trends (black, cyan curves) are so linear. Were they computed from daily model output ?*

All time evolution fields were computed from 3 days averaged outputs. They consist of the averaged quantities within the eddy, from 100-400 m depth (in the new version from 100-200 m depth- following the referee 2 suggestion). The cumulative quantities yield an almost linear curve because of the scale used in the plots. This is now rectified in the new manuscript.

*-The quality of Figure 7 needs to be improved. Labels are deformed, it is difficult to read the dates on the x axis. Also the cumulative nitrate and nitrite consumption appear as positive values, which is misleading. How was the eddy volume defined? Which criteria was used for the subsurface?*

The labels in Figure 7 and Figure 10 were changed. The cumulative nitrate and nitrite are positive, as they reflect the incremented nitrate over time. The nitrate and nitrite consumption are now negative in the new manuscript. The eddy volume is defined by the eddy area and the surface-400 m thickness. Following the referee 2 comment, the eddy volume is now defined as the eddy area and the 100-200 m thickness in the revised manuscript,

*P5,L23 : I do not see anything at the edges of the eddies in Fig 8c. Are you referring to the tiny peaks near 200-250m depth ? Is that relevant ?*

Indeed, there is no significant nitrate production and reduction at the edge of the eddy. The sentence is removed. We apologize for this error.

*Fig9ab: what happened at the end of january ? What explains these nitrate/nitrite peak?*

The nutrients peak end of January might be related to enhanced supply by the lateral transport at the edge of the eddy.

Following the comments of Referee 2, the eddy volume is now reduced to 100-200 m and the simulated dynamics have slightly been modified. As consequence the above comment is no longer applicable for the analysis in the new manuscript.

**P6,L2 : I am not sure I clearly understand what is meant here. Do you mean eddies capture surrounding waters in their core when they are formed, and then propagate with the trapped water mass ? When the eddy has been formed, the surrounding waters are entrained on the edges of the eddy, which creates horizontal stirring.**

I meant that eddies can exchange the water masses with the surrounding environment during their propagation.

Indeed, after the eddy formation, the surrounding waters are entrained at the edge of the structure. Part of this entrained waters can enters the eddy interior, contributing to the production within the eddy.

**P5,L3 : could you explain the process here ?**

Changed accordingly.

**Addition to the text, section 2.2.2, page 5:**
The warmer and saltier waters within the mode water eddies is  related to the depression of the  lower isotherms and isohalines in the interior of the structure.

**P6,L10 : I did not understand where Asim was formed. Is it further north or south? The eddy is still quite close to the coast. It would be really useful to add a figure which clearly shows the trajectories of the two eddies, since you study their temporal evolution.**

The formation and trajectory of the eddy $A_{sim}$ and $B_{sim}$ are now shown in the new

manuscript.

*Figure 10c : the colors are a bit confusing, as the y axis of the TS diagram already indicates the temperature of the WM. The magenta points in Fig 10c suggest that the eddy was formed further north, away from the area as its original WM differ from the surrounding WM.*

The magenta points indicate that the water masses present within the eddy during formation are warmer than the water mass present at the instance shown in Figure 4. To clarify the Figure 10, the spatial distribution of temperature and salinity an instant following the eddy formation is presented in the supplementary material (Fig.SI-3 and Fig.SI-4 ) and discussed in the new version of the manuscript.

*Fig 11 : would be easier to read if there was a zoom on the eddy. No need to show the whole domain. The eddy is far from the coast.*

Changed accordingly.

*Fig11ef : I do not see saltier water entering the eddy, but rather fresher, slightly cooler and nutrient richer water entering the eddy in Fig11e-f-h.*

*Changed accordingly.*

**Addition to the text, section 3.2, page 8:**
At 100 m   depth, warmer and saltier southern waters occupy the eddy interior (Fig. 13-e,f,g). These waters also dominate the eddy interior at the instant following the eddy formation (Fig. SI4-e-h in supplementary information). The waters of southern origin that are   trapped within the eddy's interior are subsequently mixed with the fresher and colder northern waters, which enters the eddy by the north-western edge.

*P6,L31 : I do not see the exchange of waters with the environment in Fig 11k. Please*

*explain.*

The water masses at the identification instant do not differ much from the water masses present within the eddy a month earlier. Else, the nutrient advective fluxes calculated at the edge for the eddy $B_{sim}$ (Fig. 10 and Fig. 12) show exchange with the surrounding environment, which results in variations in the nutrient distribution within the eddy structure (Fig. 10 and Fig. 12).

*P7,L1-2 : These lines are not convincing. You say that there is exchange, then that it is not strong. I do not see clearly where you are going with that.*

Those lines are removed in the new version.

*Conclusions section :*
*P7,L10 : I think that it is not clear why the WM are different when the eddies are formed. Is it the location ? The season ? An interesting and possibly more convincing diagnostic would be to show the nitrate/nitrite concentration in the newly formed eddies in the same*

Nitrate and nitrite concentrations at the instant following the eddy formation are now shown in the supplementary information (Fig. SI-3 and Fig. SI-4) and discussed in the new version of the manuscript.

*Figure. Also I don't understand how you can compare these 2 eddies, which obviously have different ages. Bsim is only 2 months old according to Figure 9, and is located very far from the coast. It seems to me that it should be older than Asim which is 3.5 months old (according to Fig 7) and closer to the coast. How could the two eddies be in such different places with such age difference ? This needs clarification.*

Indeed, the two selected eddies are not located in the same region. The choice of the eddies (location mainly) was guided by the eddies observed by Stramma et. al. (2013) off Peru. The high nitrate eddy was located as in Stramma et. al. (2013) offshore. The low nitrate

eddy was located close to the shelf. We added the respective information to the revised version of the manuscript.

*P17,L12-13 : This may be true but it remains to be demonstrated based on dedicated diagnostics.*

We are thankful to the referee 3 for pointing out the need for a more detailed diagnostics to support the conclusion of this study. To respond to this comment, we present in the new manuscript advective fluxes of both nitrate and nitrite during the eddy propagation. The nutrient fluxes across the eddy's limits are higher than the biogeochemical rates of nutrient production. This support our findings that the nutrient patterns within the eddies are driven by physical dynamics rather than by local on-going biogeochemical processes.

**Addition to the text, section 3.2, page 7 and  page 8:**

…. To investigate the origin of water masses present in the selected eddies, we analyse the advective transports of both nitrate and nitrite into the eddy during the eddy's lifetime (Fig. 10 and Fig. 12). The water mass properties within the structure are also analysed and compared with the surrounding environment during different instants of the eddy's lifetime (Fig. 11 and Fig. 13). Figure 10 illustrates the nitrate and nitrite fluxes into the eddy $A_{sim.}$ It shows a strong injection of nutrients from the lateral margins of the eddy. This nutrient injection is elevated in the first months following the eddy formation. The cumulative fluxes of both nitrate and nitrite significantly increase in this period and follow the evolution of both nitrate and nitrite within the eddy. These dynamics suggest a strong exchange with the surrounding environment during this period. This is also visible in the water mass properties within the eddy structure (Fig. 11). At the surface, waters present within the eddy $A_{sim}$ are relatively cool and fresh compared to the water masses present following the eddy formation (Fig. 11-a-b)…..

The nutrient fluxes across the edge of the eddy $B_{sim}$ are presented in Figure 12. It shows a contribution of both horizontal and vertical transport to the nutrient variation within the eddy, during the eddy's lifetime. After the eddy $B_{sim}$ formation, the nitrate fluxes through the

edge of the eddy B$_{sim}$ are dominantly out-going, showing a loss of nitrate to the surrounding environment (Fig. 12-a). These out-going fluxes reduce the nitrate availability within the eddy. About half a month later, the nitrate concentration within the eddy increases. This increase is to a large extent due to the nitrate supply into the eddy structure from both vertical and horizontal boundaries. On the contrary, the nitrite supply into the eddy is largest and positive in the month following the eddy formation and decreases afterwards (Fig. 12-b).

[Figure]

**Figure 10.** Nitrate (a) and nitrite (b) advective fluxes into the eddy A$_{sim}$. Lines indicate horizontal (solid blue, µmol l$^{-1}$d$^{-1}$), vertical (dashed blue, µmol l$^{-1}$d$^{-1}$) and cumulative (black, µmol l$^{-1}$) advection. Red line represents the available nitrate within the eddy [µmol l$^{-1}$]. Arrows indicate the time where the sections in Figure 4 were taken.

[Figure]

**Figure 12.** Nitrate (a) and nitrite (b) advective fluxes into the eddy B$_{sim}$. Lines indicate horizontal (solid blue, μmol l$^{-1}$d$^{-1}$), vertical (dashed blue, μmol l$^{-1}$d$^{-1}$) and cumulative (black, μmol l$^{-1}$) advective fluxes. Red line represents the available nitrate within the eddy [μmol l$^{-1}$]. Arrows indicate the time where the sections in Figure 5 were taken.

*I also do not think that you can base your conclusions on the examination of only two eddies. There must be plenty of eddies in your multi-annual simulation, from which you can compute some more robust statistics.*

One counter-example can render a theory obsolete or even prove it wrong. Our aim here is to show that biogeochemical cycling in eddies similar to the ones observed by Stramma et al. (2013) may be governed by processes other than the one suggested, based on an

observed snapshot, by Altabet et al. (2012) and Stramma et al. (2013). A more comprehensive eddy census is beyond the scope of the manuscript.

Nonetheless, in order to gain more insight into the processes governing the nutrient dynamics within anticyclonic eddies, we conducted a particle-release experiments. Anticyclonic eddies tracked at the vicinity of the particle-release locations have trapped particles within their centre. The number of particles within these eddies varied in time, suggesting an exchange with surrounding environment during their propagation. This result supports the hypothesis that nutrient patterns within anticyclonic eddies can be strongly affected by physical exchange processes with surrounding waters.

**Addition to the text, section 2.4, page 5 and page 6:**

**2.4 Particle release experiment**

In order to have a more general overview of the processes controlling the dynamics of the eddies in the ETSP, we conduct a particle-release experiments and analyse the anticyclonic eddies that are in the vicinity of the particle-release locations at the time of the release. In these experiments, particles are released in three different locations in ETSP: (1) along the shelf between $13^o$S - $15^o$S, (2) along the shelf between $9^o$S $11^o$S, and (3) offshore between $13^o$S-$15^o$S in latitude and $85^o$W-$86^o$W (cf. Fig. SI-2 in supplementary information). The particles are released in the entire water column on the shelf and in the upper 300 m at the offshore site, in early austral summer (January) and early winter (June) of the last three climatological years of the model simulation.

**Addition to the text, section 3, page 8 and 9:**

**3.3 Eddy stirring and nutrient entrainment**

The eddies $A_{sim}$ and $B_{sim}$ showcase that the nutrient supply by physical dynamics is the dominant mechanism that controls simulated (diverse) nutrient pattern within the eddies. The nutrient exchange with surrounding waters occurs throughout the entire lifetime of the eddies.. This indicates that the nutrient availability in the vicinity of the eddy plays a role for the nutrient distribution within the eddy's structure. To elucidate this suggestion, we carried out particle-release experiments (subsection 2.4) and analysed the eddies that passed and/or

were generated close to the particle-released areas. Figure 14 illustrates the particle distribution in the subsurface interior (between 100 - 200m depth) of anticyclonic eddies during their propagation. From the early stages on, particles are entrained and trapped within the eddy structures.

[Figure]

**Figure 14.** Particle distribution at the subsurface layer of anticyclonic eddies during their propagation, during summer (left) and winter (right) seasons. Anticyclonic eddies tracked at the vicinity of southern (a,b), northern (c,d) and offshore (e,f) particle-release locations. Detailed description of particle-release experiments can be found in the supplementary information.

These particles are transported offshore during propagation of the eddies. Every tracked

eddy shows a pronounced temporal variation of the amount of particles within the structure, an indicative of exchange of properties with surrounding waters. This behavior occurs in eddies tracked during both austral summer (Fig. 14-a,c,e ) and austral winter (Fig. 14-b,d,f ).

**Addition to the text, section 4, page 10:**

Anticyclonic eddies tracked during the particle-release experiments corroborate this suggestion and show the occurrence of water mass exchange between the eddy and the surrounding environment. Particle numbers within these eddies are repeatedly increased and decreased, showing a loss and gain of quantities to/from the surrounding environment.

**Addition to the text, section 5, page 10:**

In a more general context the particle-release experiments realized in this study also emphasize the role of water mass exchange between eddies and the surrounding environment for the temporal evolution of properties within the eddy structure.

**Addition to the supplementary information, section 3, page 2:**

**3. Particle-release experiments**

In Figure SI-2 are illustrated the locations of particle release (light blue) as well as the tracked anticyclonic eddies (filled circles and triangles) during the model particle-release experiments. The particle-release experiments, which consisted of releasing inactive Lagrangian particles along the shelf and off Peru, are conducted to investigate the capability of anticyclonic eddies to exchange water masses with the surrounding environment. In order to cover possible seasonality effects, the particles were released in both summer and winter seasons of the southern hemisphere.

[Figure]

**blue for year 50. Eddies tracked during summer and winter are represented in filled circles and triangles respectively.**

*P17,L14 : Weakened flow relative to what ?*

Sentence removed from the conclusion and rewritten in the actual discussion section

*You seem to imply the WM contained in the eddy depend on the formation mechanism or site, but this is not clear.*

We apologize for the lack of clarity regarding the processes affecting the nutrient distribution within the eddy. In order to clarify the results obtained by analysing the water masses, the analysis of the advective nutrient transport are now added to the revised manuscript. Further, the results are discussed in a separate section, which allows for a more comprehensive explanation of the processes occurring within the eddy.

---

## Author Comment (AC4) · 25 Oct 2016

[revised manuscript text omitted]

---

## Author Response (AR2)

**Detailed response to the reviewer**

First of all, we would like to thank the reviewer for her/his constructive comments, which have allowed further clarification of the manuscript. We have addressed each of the concerns as outlined below.

**Specific Comments from the Reviewer**

*P4 L3 "However, the magnitude of simulated nitrite is lower compared to the observations."*
*Is this sentence for the inshore nutrients?*

Yes, the underestimated nitrite is on the inshore. Sentence is now clarified, as follow:

**Addition to the text, section 2.1, page 4, line 2:**
However, the magnitude of simulated inner-shelf nitrite is lower compared to the observations

*P5 L7 "Despite reproducing the subsurface nitrite maxima, the simulated concentrations are an order of magnitude lower compared to the concentration within the observed eddy"*
*Which observation is this? Is it in-situ observation?*

Yes, the mentioned observed eddy correspond to an in-situ observations. Sentence is now clarified as follow:

**Addition to the text, section 2.2.2, page 5, line 7-8:**
Despite reproducing the subsurface nitrite maxima, the simulated concentrations are an order of magnitude lower compared to the in situ observations of nitrite concentration within eddy (Stramma et al, 2013).

*P6 L6 "and and"*
*Remove one of two "and".*

Changed accordingly.

*P6 L29 "These results show that the low nitrate (high nitrite) within the eddy Asim is not primarily controlled by local biogeochemical dynamics within the eddy Asim"*
*What missing in the presented analysis in Figure 7 and 9 is that there is no information about the time evolution of the eddy volume. In addition, as the eddy travels, some portions of the water can be transported as the eddy flux depending on the nonlinearity i.e., u/c, where u is the current velocity of an eddy and c traveling speed of an eddy, and some water is not. This is problematic. If the volume of water, which the authors tracked for the analysis in Figure 7 and 9 is all trapped within the eddy, all variations in tracers must be controlled by the local processes. In contrast, if the eddy is a completely linear wave, and no water is fluxed by the eddy, the biogeochemical*

*processes within the tracked eddy (wave) is all nonlocal processes. Instead, it is just looking at different water column as the wave propagates. In reality, as authors pointed, some water is held inside the eddy and some not. In summary, it is very difficult to say if it is caused by the local processes or not, if the volume of the water is varied as the eddy evolves.*

First, we would like to clarify the term local processes. In this study the local processes refer to the biogeochemical processes (which are the source-sink terms) occurring within the eddy structure. Indeed, if the eddy does not exchange water properties with the surroundings, the tracers quantities will be affected only by the local processes. This is not the case for the analysed eddies. The water masses as well as the tracer fluxes show some exchange of water with surrounding environment while the eddy moves. This gives an indication that remote processes can influence the tracer dynamics within the eddy.

Further, the eddy volume varies as the eddy moves, mainly due to changes in horizontal structure of the eddy. This variation of the eddy volume is accounted while calculating the contribution of the local processes on a concentration basis, i.e. per cubic meter. We admit that there is no perfect way of describing local and remote processes in a changing water volume. We thank the reviewer for the suggestion to show the variations in the eddy volume, which gives the reader a chance to better understand what is going on.

The variation of the eddy volume is now presented in the supplementary information.

We have added to the section 2.3, line 23 " (Fig. SI-2 in supplementary information for further details)."

**Figure 7 Caption "Bsim" should be "Asim".**
Changed accordingly.

**P7 L13 "In the eastern tropical ocean, eddies have been observed to carry low-oxygen waters out of the core region of the OMZ"**
**Is this applicable for only anticyclonic eddies or both cyclonic and anticyclonic eddies?**

So far and to our knowledge, observations only show anticlyclonic eddies carrying low-oxygen waters out of the core region of the OMZ. This is now clarified.

**Addition to the text, section 3.2, page 7, line 9:**
In the eastern tropical oceans, anticyclonic eddies have been observed with low-oxygen waters out of the core region of the OMZ.

**Figure 10 and 12, and related text**
**The unit of the flux is not for flux. It is flux divergence.**
**Similar to the comment for Figure 7 and 9, time variation of the volume tracked is needed.**

Changed accordingly. The variation of the eddy volume is now presented in the supplementary information.

We have added to the section 2.3, line 23 " (Fig. SI-2 in supplementary information for further

details)."

*Figure 11 and 13 and related text*
*These results are rather descriptive hydrography in the numerical model, and sounds more like discussion as the authors speculate the importance of the advection just by the T-S diagrams. If water properties are horizontally (along isopycnally) uniform, T-S properties are unchanged by the lateral (along isopycnall) advection. These water exchanges could be addressed more quantitatively by the Lagrangian particle tracking.*

The Lagrangian particle tracking  is presented in the manuscript, section 3.3.

*P8 L20 "(Fig. 11-e-f).n)"*
*There is no panel n.*

Sorry about this typo. This is now removed.

*P9 L7 "Subsurface anticyclonic eddies…"*
*I did not realized a fact that this study focused on the subsurface anticyclonic eddies. If so, these previous studies for mode water eddies or under current eddies should be in the introduction. If the study is focused on the subsurface eddies, the detection method needs to be modified for them.*

The study focus on the mode water eddies. Altabet et al. (2012) and Stramma et al. (2013) are mentioned in the  introdution, on page 2, line 2-5.

We do not agree with the reviewer comment on the need of change the detection method. As for the observed mode water eddies in the region off Peru and tropical North Atlantic Ocean, the  simulated mode water eddies present a surface signal on the sea surface height. This characteristic allows a detection by the method applied in our study.

*P9 L27 "similar the observed" should be "similar to the observed".*

Changed accordingly.

*P8 3.3 Eddy stirring and nutrient entrainment*
*It is good that the authors conducted a Lagrangian simulations. The results would show importance of the water exchanges, but are not clear. This is because these entrained particles do not necessarily represent the main body of the water inside the eddy. It is better if the particles are back tracked from the inside eddy with different depths. This is more direct to know where the water comes from.*

The back tracking of particles from the inside eddy would indeed tell us the origin of the particles present at a given time during the eddy propagation. This would, however, not tell much about the exchange with the surrounding environment during the eddy propagation, which is one of the themes of that subsection. By analysing the particle dynamics during the eddy propagation, we could show

that the eddies in this region exchange properties with the surrounding environment and that this contributes to the dynamics within the structure. Else, it also emphasizes that the concentration of a given tracer at the vicinity of an eddy structure can substantially affect the tracer's concentration within the eddy.

***Figure 14b includes the number of particles inside Asim and shows that it is not largely changing compared to that of other eddies. This is not consistent with the claim that there is strong advective inputs of nitrate into Asim.***

In the first month, the eddy $A_{sim}$ have lost around $40.10^6$ particles/liter, a number much higher compared to the other eddies. We would like to call your attention to the scale variation in the Figure 14b.

We have added to the figure caption "Note the change of the scale on the vertical axes."

***P9 L27-33 "In the low nitrate (high nitrite) Asim eddy (Fig. 4-c), the on-going nitrate reduction by denitrification is lower than the nitrate production by nitrification. This fails to explain the low subsurface nitrate within the eddy, which is in contrast to the interpretation of the observations by Altabet et al. (2012) and Stramma et al. (2013). Further, we find in our simulation that the advective fluxes across the edge of the eddy Asim shows a strong nitrate supply into the eddy, in the first days prior to the eddy formation. This nitrate supply, which is predominately horizontal, is a consequence of the exchange of water masses with the surrounding environment during the eddy propagation."***

***I'm confused by these sentences. The results indicate that local biogeochemical processes, which imply increase in nitrate, and the advective flux divergence also results in a net increase in nitrate. So both estimates cannot explain the low nitrate inside the eddy?***

The reviewer is correct, both local processes and advective fluxes of eddy $A_{sim}$ show a net increase of nitrate. This shows the complexity and points out that the origin of supplied water is of importance when interpreting eddy dynamics. At the instant following the eddy formation both shelf and offshore waters are present within the subsurface layer of the eddy interior. However, the offshore waters, depleted in both nitrate and nitrite, dominate the water masse properties. During the eddy propagation, shelf waters dominate the water masses within the eddy interior, as shown in the TS diagram. Shelf waters have relatively high nitrate concentrations and are rich in nitrite in comparison to the offshore environment. This results in the net nitrate (nitrite) supply during the eddy propagation.

This analysis is now clarified in the manuscript. To simplify the analysis, the Figure 11-i-l and 12-i-l (as well as in the supplementary information) show temperature, salinity and tracers at 200 m depth, which is the lower limit of the eddy volume instead of 250 m depth as previously shown.

**Addition to the text, section 3.2, page 7, line 32-34 and page 8, line 1-5:**
Further down (at 200 m depth), warmer and saltier shelf waters dominate the properties within the eddy $A_{sim}$ (Fig. 11-i,j). At the eddy formation, both shelf and offshore waters occupy the eddy interior (Fig. SI-3-i-k). The surrounding offshore waters have higher concentrations of both nitrate and nitrite (Fig. SI-3-k,l). With the eddy propagation, water properties are modified (Fig. 11-k). Shelf waters, which have lower nitrate concentrations compared to the offshore environment, are kept in

the eddy centre and advected during the eddy propagation (Fig. 11-l). Thus, the nitrate concentrations in the eddy interior are lower than those of the surrounding waters, even though interior concentrations show an increase over time during the propagation of the eddy.

*P10 L27- "The model results show a decoupling between local nitrate reduction (nitrite production) via biogeochemical processes and total changes in nitrate (nitrite) within the eddy"*
*As mentioned in my comments above, it is difficult to separate local and advective processes, because how much water is trapped affects the locality with respect to the moving eddy.*

We are sorry about the confusion. This, as well as the comment to P6L29 is the consequence of us having failed to define "local processes". We have done this now in the revised version of the manuscript: "Local processes" refer to local, on-going biogeochemical sources and sinks (as opposed to the effects of divergent physical transport processes or initial conditions)

*As it is clear in the conclusion, the results of this study could not successfully close the budget. The both local and advective processes do not explain the nitrate concentration within the eddy quantitatively.*

The aim of the present study is to understand the processes effecting nitrate patterns within the eddy. To address this objective, we looked at different processes, which had been suggested previously. We could demonstrate that local biogeochemical processes alone cannot quantitatively explain the observed patterns. We then analysed both water mass properties and advective processes. This analysis has shown the complexity of dynamics within the eddies, where many processes impact on the tracer's distribution. We could see that (1) the conditions at the place of eddy formation imprint a long-lasting nutrient signature within the eddy. (2) These initial conditions are modified by physical transport processes occurring at the edge of the structure. The contribution of advective processes on the nutrient signal within the eddy was quantified.

These results have been supported by Lagrangian particle tracking. If we take as example the eddy $A_{sim}$, which was formed close to the source of particles release, it exhibits a high amount of particles within its core. When the eddy moves out of the particle-release source, the number of particles declines. This is mainly due to exchange with the surrounding waters poor in particles.

The conclusion of the present study was, therefore, based on these three elements (1) local processes, (2) origin of trapped waters and (3) advective processes.

*P11 L1 "shaping the nutrient patterns within cyclonic eddies"*
*Where are the results for cyclonic eddies?*

We were suggesting that the water mass exchange with surrounding environment might also influence the nutrient pattern within cyclonic eddies, even though we do not yet have any direct evidence for this claim. This sentence is now removed from the new version of the manuscript.

---

## Author Response (AR3)

Dear Dr Bopp

First of all, we would like to thank you for your support and for giving us the opportunity to resubmit revision of this manuscript.

We have addressed all issues indicated in the referee's comments and this is now uploaded for further reviewing process.

Yours sincerely

Yonss José (on behalf of the co-authors)

**Detailed response to the referee #1**

We would like to thank referee 1 for the constructive comments, which have allowed a considerable improvement of the manuscript. The most substantial point the referee stated concerned the analyses of the advective fluxes of nutrients. The respective analysis has been added to the revised version of the manuscript and it further supports our conclusions.

**General Comments**

This paper shows results from a coupled physical and biogeochemical ocean numerical simulation. The authors identified two anticyclonic eddies off Peru in the simulation and showed that the local biogeochemical processes in the model can not explain the temporal variations in nitrate concentration within eddies. Although the authors found that local biogeochemical processes are less important than advective processes, the analyses presented by the authors were mostly focused on the role of biogeochemical processes, with some hydrographic investigations to track the origin of the advected water. Because the authors found that advective processes are important, then the advective fluxes should be analyzed directly, as they are available from the numerical results. Another effective tools to track the origin of the water mass would be particle tracking. Without these analyses, the paper seems to lack the supporting evidences to derive the conclusions that nutrient injected into and out of the eddies.

We agree with Referee #1 that models offer the opportunity to perform in-depth analysis of advective processes and that this will bring more insight on the processes playing a role on nutrient variation within the eddies. Following Referee #1's advice we have computed the advective fluxes within the selected eddies. We focus on the vertical and horizontal advection because in the high-resolution configuration used here, they are of leading order (Fig.1 and Fig. 2).

**Figure 1.** Nitrate (upper panel) and nitrite (lower panel) advective and diffusive fluxes within the eddy  $A_{sim}$ . Lines indicate horizontal (black) and vertical (blue) advection [µmol l-1d-1], horizontal (red) and vertical diffusion [µmol l-1d-1]. Note the different scales on the left (advective transport) and right (diffusive transport) axis.